# IGNIS: A Robust Neural Network Framework for Constrained Parameter Estimation in Archimedean Copulas

## Abstract

Classical estimators, the cornerstones of statistical inference, face insurmountable challenges when applied to important emerging classes of Archimedean copulas. These models exhibit pathological properties, including numerically unstable densities, a restrictive lower bound on Kendall's tau, and vanishingly small likelihood gradients, rendering methods like Maximum Likelihood (MLE) and Method of Moments (MoM) inconsistent or computationally infeasible. We introduce **IGNIS**, a unified neural estimation framework that sidesteps these barriers by learning a direct, robust mapping from data-driven dependency measures to the underlying copula parameter $\theta$. IGNIS utilizes a multi-input architecture and a theory-guided output layer $(\mathrm{softplus}(z) + 1)$ to automatically enforce the domain constraint $\hat{\theta} \geq 1$. Trained and validated on four families (Gumbel, Joe, and the numerically challenging A1/A2), IGNIS delivers accurate and stable estimates for real-world financial and health datasets, demonstrating its necessity for reliable inference in modern, complex dependence models where traditional methods fail.

## 1 Introduction

Maximum Likelihood Estimation (MLE), a pillar of statistical inference, is the gold standard for parameter estimation due to its desirable asymptotic properties. Its efficacy, however, is predicated on well-behaved likelihood functions. In the domain of dependence modeling using copulas (Nelsen, 2006), this assumption can dramatically fail. For a growing class of flexible and important models, such as the novel A1 and A2 Archimedean copulas (Aich et al., 2025), the likelihood function exhibits pathological properties that render classical estimation methods inconsistent, unstable, or computationally infeasible. This issue is not isolated; numerical challenges in copula estimation are a known and significant concern in high-stakes applications like quantitative risk management (Hofert et al., 2013).

Our analysis of these challenging models reveals three fundamental barriers that make classical estimation untenable:

1. **Numerical Instability from Boundary Singularities:** The copula density function, which is required for MLE, explodes near the boundaries of the unit hypercube due to ill-behaved generator derivatives (e.g., with singularities of order $O(t^{-3})$), leading to floating-point overflow during computation.

2. **Inapplicable Dependence Range:** For MoM to be viable, a model's theoretical range of dependence must cover the empirical dependence of the data. The A1 and A2 families are severely limited in this regard, as their range for Kendall's $\tau$ begins at approximately 0.54518, making them incapable of modeling the weak or moderate dependence prevalent in many real-world datasets.

3. **Vanishing Gradients and Hessian Decay:** For even moderately large values of $\theta$, the log-likelihood surface becomes pathologically flat. The score function decays polynomially to zero (e.g.,

$O(\theta^{-8})$), causing gradient-based optimizers to stall prematurely. Second-order information decays even faster, rendering Newton-like methods useless.

Recent deep learning approaches have shown immense promise in statistics, but have not addressed this specific estimation problem. The state-of-the-art has largely focused on generative tasks, such as learning new copula generators from scratch (Ling et al., 2020; Ng et al., 2021) or modeling highly complex, high-dimensional dependence structures (Ng et al., 2022). However, the fundamental problem of robust parameter estimation for known, specified families that exhibit the aforementioned pathologies remains a critical open gap. To fill this gap, we introduce **IGNIS**, a unified neural estimation framework that sidesteps the pitfalls of classical methods entirely.

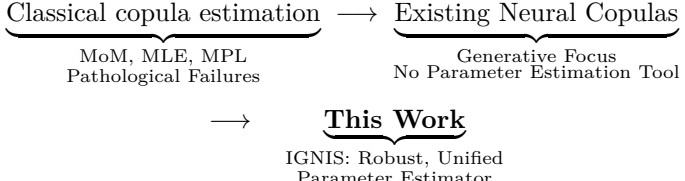

IGNIS learns a direct mapping from a vector of robust, data-driven summary statistics to the underlying copula parameter $\theta$. Our main contributions are:

1. The identification and formal analysis of three critical optimization barriers that cause classical estimators to fail for an important class of copula models.

2. The design and implementation of IGNIS, a unified neural architecture that learns a robust estimation function and enforces theoretical parameter constraints ($\hat{\theta} \geq 1$) via a custom output layer.

3. A comprehensive validation on simulated and real-world data, demonstrating that IGNIS provides accurate and stable estimates precisely in the regimes where traditional methods break down.

The remainder of this paper is organized as follows. Section 2 reviews related work. Section 3 presents the notations used in the paper. Section 4 presents necessary preliminaries. Section 5 provides motivation for our work. Section 6 details the IGNIS architecture and training protocol. Section 7 presents the simulation results for IGNIS, and Section 8 demonstrates real-data applications. Finally, Section 9 concludes and outlines future research directions.

## 2 Related Work

Our work builds upon two distinct streams of literature: classical parameter estimation for copulas and the emerging field of deep learning for statistical modeling.

### 2.1 Classical Estimation and its Limitations

Parameter estimation for Archimedean copulas has traditionally been approached via two main routes. The Method of Moments (MoM), particularly using Kendall's $\tau$ or Spearman's $\rho$, is valued for its computational simplicity and circumvention of the likelihood function (Genest & Rivest, 1993). However, both A1 and A2 have a **high lower bound** for Kendall's $\tau$ ($8\ln 2 - 5 \approx 0.54518$), which makes MoM inapplicable to many real datasets with weaker dependence.

The second route is Maximum Likelihood Estimation (MLE) or its semi-parametric variant, Maximum Pseudo-Likelihood (MPL) (Genest et al., 1995). While asymptotically efficient, MLE requires computing the copula density, which can be analytically complex and numerically unstable. Efforts by (Hofert et al., 2011) derived explicit generator derivatives to make MLE more feasible for standard families. Yet, subsequent large-scale studies confirmed that even with these advances, classical estimators face significant numerical

challenges and potential unreliability, especially in high dimensions or for complex models (Hofert et al., 2013). The A1 and A2 families are prime examples where these numerical pathologies become insurmountable barriers, necessitating a new approach.

### 2.2 Deep Learning Approaches to Copula Modeling

The recent intersection of deep learning and copula modeling has been dominated by powerful generative approaches that learn or approximate the generator function itself, rather than estimating parameters of a pre-defined family. For instance, ACNet (Ling et al., 2020) introduced a neural architecture to learn completely monotone generator functions, enabling the approximation of existing copulas and the creation of new ones. Similarly, (Ng et al., 2021) proposed a generative technique using latent variables and Laplace transforms to represent Archimedean generators, scaling to high dimensions. Other work has focused on non-parametric inference for more flexible classes like Archimax copulas, which are designed to model both bulk and tail dependencies (Ng et al., 2022).

While these methods represent the state-of-the-art in constructing flexible, high-dimensional dependence models, they do not address the targeted problem of estimating the parameter $\theta$ for a specified family, especially when that family exhibits the estimation pathologies we have identified. Broader work on Physics-Informed Neural Networks (PINNs) has shown the power of deep learning for solving problems with known physical constraints (Raissi et al., 2019; Sirignano & Spiliopoulos, 2018), but a specialized framework for constrained parameter estimation in statistically challenging copula models has been a missing piece. IGNIS is designed specifically to fill this gap, providing a discriminative estimator that is robust, constraint-aware, and applicable across multiple families where classical methods fail.

## 3 Notation

Throughout our analysis, we employ a consistent set of symbols. The core parameter of an Archimedean copula is denoted by $\theta \in [1, \infty)$, with its estimate from our framework being $\hat{\theta}$. The copula function itself is $C(u, v)$, constructed via a generator function, $\phi(t)$, and its inverse, $\phi^{-1}(s)$. To ensure clarity, we distinguish this from its corresponding probability density function, $c(u, v)$. In theoretical contexts (Appendix C), the standalone uppercase letter $C$ denotes the copula family, while subscripted variants (e.g., $C_k$) represent constants within proofs. For the Method of Moments, we use the theoretical Kendall's tau, denoted by $\tau$.

Our neural network framework, IGNIS, is trained on a dataset of $N$ examples. Each example is an input vector $\mathbf{x} \in \mathbb{R}^9$. This vector is a concatenation of two components: a 5-dimensional vector of continuous summary features, $\mathbf{f} \in \mathbb{R}^5$, and a 4-dimensional one-hot vector indicating the copula family, $\mathbf{c} \in \{0, 1\}^4$. The feature vector $\mathbf{f}$ is comprised of five empirical dependency measures calculated from a data sample of size $n$: Kendall's tau ($\tau_n$), Spearman's rho ($\rho_n$), the Pearson correlation coefficient ($r_n$), and coefficients of upper ($\lambda_{upper,n}$) and lower ($\lambda_{lower,n}$) tail dependence.

The neural network has $D$ layers and is trained to minimize a mean squared error loss function $\mathcal{L}(\theta)$ by adjusting its weights and biases using the Adam optimizer with a learning rate $\eta$. For the theoretical consistency proof presented in Appendix C, the feature vector is denoted by $\mathbf{T}_n$, and the set of all possible feature vectors is the feature space $\mathcal{T}$.

## 4 Preliminaries

### 4.1 Copulas and Dependency Modeling

Copulas are statistical tools that model dependency structures between random variables, independent of their marginal distributions. Introduced by Sklar (1959), they provide a unified approach to capturing joint dependencies. Archimedean copulas, known for their simplicity and flexibility, are defined using a generator function, making them particularly effective for modeling bivariate and multivariate dependencies.

### 4.2 The A1 and A2 Copulas

Like all Archimedean copulas, the novel A1 and A2 copulas (Aich et al., 2025) are defined through generator functions $\phi(t)$ that are continuous, strictly decreasing, and convex on $[0, 1]$, with $\phi(1) = 0$. The A1 and A2 copulas extend the Archimedean copula framework to capture both upper and lower tail dependencies more effectively. In general, an Archimedean copula is given by:

$$C(u, v) = \phi^{-1}\big(\phi(u) + \phi(v)\big). \tag{1}$$

For the A1 copula, the generator and its inverse are defined as:

$$\phi_{A1}(t; \theta) = \left(t^{1/\theta} + t^{-1/\theta} - 2\right)^{\theta}, \quad \theta \geq 1, \tag{2}$$

$$\phi_{A1}^{-1}(t; \theta) = \left[\frac{t^{1/\theta} + 2 - \sqrt{\left(t^{1/\theta} + 2\right)^2 - 4}}{2}\right]^{\theta}, \quad \theta \geq 1. \tag{3}$$

Similarly, for the A2 copula:

$$\phi_{A2}(t; \theta) = \left(\frac{1-t}{t}\right)^{\theta}(1-t)^{\theta}, \quad \theta \geq 1, \tag{4}$$

$$\phi_{A2}^{-1}(t; \theta) = \frac{t^{1/\theta} + 2 - \sqrt{\left(t^{1/\theta} + 2\right)^2 - 4}}{2}, \quad \theta \geq 1. \tag{5}$$

The exact formula of the Kendall's $\tau$ for A1 and A2 copulas are given by (See Appendix A for full derivations)

$$\tau_{A1} = 3 + 4\theta\left[\psi(\theta) - \psi\left(\theta + \frac{1}{2}\right)\right], \tag{6}$$

$$\tau_{A2} = 1 - \frac{6 - 8\ln 2}{\theta}. \tag{7}$$

While Eq. 6 is complex, it can be shown that $\tau_{A1}(\theta)$ is strictly monotone increasing on its entire domain of $\theta \geq 1$; a formal proof is provided in the Appendix A.

Both copulas are parameterized by $\theta \geq 1$, which governs the strength and nature of the dependency. The dual tail-dependence structure of A1 and A2 copulas is particularly valuable for modeling extreme co-movements in joint distributions. In financial risk management, they can capture simultaneous extreme losses (lower-tail) and windfall gains (upper-tail), improving estimates of portfolio tail risk. In anomaly detection, they identify coordinated extreme events (e.g., simultaneous sensor failures in industrial systems or cyber attacks across networks) by quantifying asymmetric tail dependencies. This flexibility makes them superior to single-tailed copulas e.g., Clayton (captures only lower tails) and Gumbel (captures only upper tail) in scenarios where both tail behaviors are critical.

### 4.3 Simulation from Archimedean Copulas

In this section, we present an algorithm introduced by Genest & Rivest (1993) to generate an observation $(u, v)$ from an Archimedean copula $C$ with generator $\phi$.

The above algorithm is a consequence of the fact that if $U$ and $V$ are uniform random variables with an Archimedean copula $C$, then $W = C(U, V)$ and $S = \frac{\phi(U)}{\phi(U) + \phi(V)}$ are independent, $S$ is uniform $(0, 1)$, and the distribution function of $W$ is $K$. In our implementation, the inverse function $K^{-1}(y)$ is computed numerically using a robust root-finding algorithm (specifically, the bisection method).

---

**Algorithm 1** Bivariate Archimedean Copula Sampling (Genest et al., 1993)

---

**Require:** Generator $\phi$, its derivative $\phi'$, inverse $\phi^{-1}$
**Ensure:** A single draw $(u, v)$ from the copula

1: Draw $s, t \overset{\text{iid}}{\sim} \text{Uniform}(0, 1)$
2: Define
$$K(x) \; = \; x \; - \; \frac{\phi(x)}{\phi'(x)}, \quad K^{-1}(y) \; = \; \sup\{\, x \mid K(x) \le y \,\}$$
3: Compute $w \leftarrow K^{-1}(t)$
4: Compute
$$u \; \leftarrow \; \phi^{-1}\big(s\,\phi(w)\big), \quad v \; \leftarrow \; \phi^{-1}\big((1-s)\,\phi(w)\big)$$
5: **return** $(u, v)$

---

### 4.4 Method of Moments Estimation

The Method of Moments (MoM) is a classical statistical technique for parameter estimation, where theoretical moments of a distribution are equated with their empirical counterparts. In the context of copula modeling, MoM is particularly advantageous when direct likelihood-based estimation is challenging due to the complexity of deriving tractable probability density functions.

In this work, we derive exact analytical formulas for Kendall's $\tau$ for both A1 and A2 copulas (see Appendix A). These formulas establish a direct relationship between Kendall's $\tau$ and the copula parameter $\theta$, allowing for robust parameter estimation. By inverting this relationship, we develop MoM estimators for $\theta$, providing a practical approach for modeling dependencies in scenarios where traditional methods like MLE and MPL may be ineffective. However, in Section 5, we see that for both A1 and A2, MoM is not efficent.

## 5 Motivation

### 5.1 Limitations in Parameter Estimation Using Method of Moments

The Method of Moments (MoM), which works by inverting a measure of dependence like Kendall's $\tau$, is a cornerstone of classical estimation. However, its use is predicated on a simple condition: the theoretical range of a copula family's $\tau$ must be able to represent the empirical $\tau$ calculated from a dataset. For many common families, this is not an issue, as their dependence range starts at or near independence ($\tau = 0$).

The A1 and A2 copulas, however, present a fundamental barrier to this approach. As derived in Appendix A, both families share the same high lower bound for Kendall's tau of 0.54518.

This high lower bound makes both families practically inapplicable for a vast number of real-world datasets that exhibit weak or moderate dependence. As shown in Table 1, the A1 and A2 copulas are significant outliers, unable to model any dependence weaker than $\tau \approx 0.54518$. Consequently, for any dataset with an empirical tau below this value, MoM estimation is not merely inaccurate, it is impossible. This motivates the need for a robust estimation framework like IGNIS that can bypass these classical limitations.

Table 1: Comparison of Theoretical Kendall's $\tau$ Ranges for Common Copula Families.

| Copula Family | Theoretical Range of Kendall's $\tau$ |
|---|:---:|
| Gumbel | $[0, 1)$ |
| Joe | $[0, 1)$ |
| **A1** | $[0.54518, 1)$ |
| **A2** | $[0.54518, 1)$ |

*Applicability.* For A1/A2, MoM based on Kendall's $\tau$ is defined only when the empirical $\tau_n$ lies in the family's range $[0.54518, 1)$; otherwise the moment equation has no solution. This restriction does not affect IGNIS.

It is also to be noted that the injectivity property of the copula generator function guarantees that each distinct value of the parameter $\theta$ produces a unique copula, ensuring the mathematical validity of the model which is true for A1 (See Appendix B).

We wish to further clarify that the fragility of MoM for the A1 and A2 family is not tied to a general notion of "high dependence," but to a specific mathematical requirement of its Kendall's $\tau$-based implementation. The method is only viable if a dataset's empirical Kendall's $\tau$ exceeds the A1 or A2 family's uniquely high theoretical lower bound of approximately 0.54518. As most standard copulas (e.g., Gumbel, Joe) can model dependence starting from $\tau = 0$, this makes the A1 and A2 copula's MoM estimator uniquely fragile and inapplicable to many real-world datasets that exhibit moderate dependence.

## 5.2 Limitations in Parameter Estimation Using Maximum Likelihood and Maximum Pseudo-Likelihood

While the Method of Moments faces fundamental limitations with A1/A2 copulas, classical likelihood-based approaches, Maximum Likelihood Estimation (MLE) and Maximum Pseudo-Likelihood (MPL), prove equally problematic due to pathological properties of the generator functions. The non-standard forms of $\phi_{A1}$ and $\phi_{A2}$ induce three critical optimization barriers (See Appendix D for full derivations).

### 5.2.1 Three Critical Optimization Barriers

**1. Numerical Instability in Density Calculations**   As $t \to 0^+$ (with $\theta$ fixed), the second derivatives of the generators blow up:

$$\left|\phi_{A1}''(t)\right| \sim O\!\left(t^{-3}\right), \qquad \left|\phi_{A2}''(t)\right| \sim O\!\left(t^{-\theta-2}\right).$$

(See Figures 1a & 1b.) Hence the copula density

$$c(u,v) = \frac{\partial^2}{\partial u \, \partial v}\, C(u,v)$$

overflows once

$$\text{A1:}\quad t < \varepsilon_{\text{mach}}^{1/3}, \qquad \text{A2:}\quad t < \varepsilon_{\text{mach}}^{1/(\theta+2)},$$

with $\varepsilon_{\text{mach}} \approx 2.22 \times 10^{-16}$.

**2. Vanishing Gradients (Score-Decay)**   As $\theta \to \infty$ (with $t \in (0,1)$ fixed), the log-likelihood score decays:

$$\left|\partial_\theta \ell(\theta)\right| = \begin{cases} O\!\left(n\,\theta^{-8}\right), & \text{A1}, \\ O\!\left(n\,\theta^{-3}\right), & \text{A2}. \end{cases}$$

Thus it falls below any fixed tolerance $\varepsilon_{\text{grad}}$ once

$$n\,\theta^{-k} < \varepsilon_{\text{grad}},$$

which for $\varepsilon_{\text{grad}} = 10^{-6}$ and $n = 1000$ yields

$$\theta_{\text{crit}}^{\text{A1}} \approx 8.2, \qquad \theta_{\text{crit}}^{\text{A2}} \approx 126.$$

(See Figures 1c & 1d.)

**3. Hessian Decay (Barrier 3)**   Again as $\theta \to \infty$ (with $t$ fixed), the scalar Hessian decays even faster:

$$\left|\partial_\theta^2 \ell(\theta)\right| = \begin{cases} O\!\left(n\,\theta^{-9}\right), & \text{A1}, \\ O\!\left(n\,\theta^{-4}\right), & \text{A2}. \end{cases}$$

In double precision ($\varepsilon_{\text{mach}} \approx 2.22 \times 10^{-16}$) this underflows once

$$n\,\theta^{-9} < \varepsilon_{\text{mach}} \implies \theta > \left(\tfrac{n}{\varepsilon_{\text{mach}}}\right)^{1/9} \approx 1.2 \times 10^2,$$

$$n\,\theta^{-4} < \varepsilon_{\text{mach}} \implies \theta > \left(\frac{n}{\varepsilon_{\text{mach}}}\right)^{1/4} \approx 4.6 \times 10^4.$$

(See Figures 1e & 1f.)

Figure 1 helps with the visualization of the three barriers.

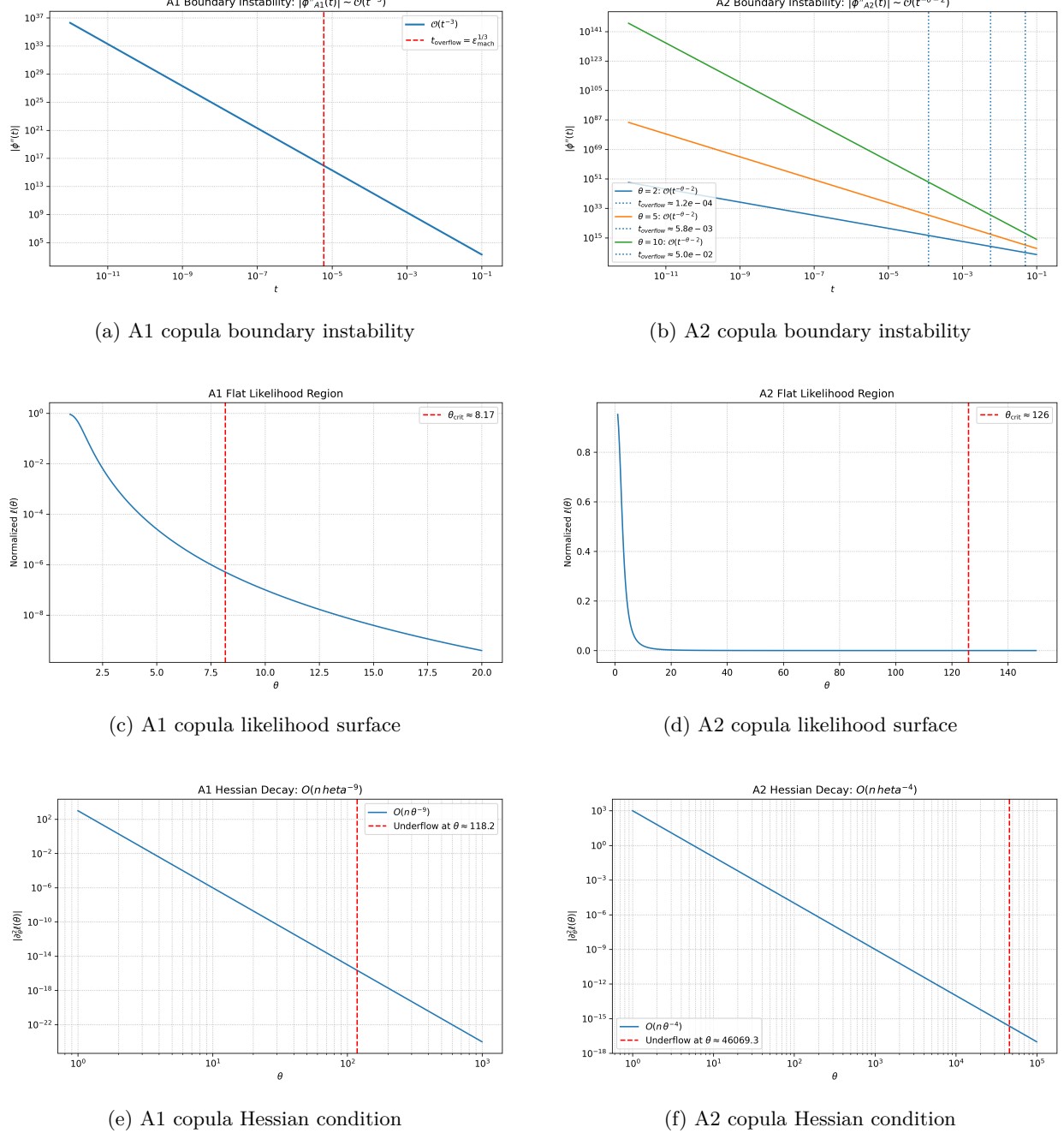

(a) A1 copula boundary instability

(b) A2 copula boundary instability

(c) A1 copula likelihood surface

(d) A2 copula likelihood surface

(e) A1 copula Hessian condition

(f) A2 copula Hessian condition

Figure 1: Numerical challenges in copula estimation: (a,b) boundary instabilities, (c,d) flat likelihood regions, and (e,f) ill-conditioned Hessian matrices for A1 and A2 copulas respectively.

Hybrid approaches, using MoM to initialize MLE/MPL, remain infeasible as well since for both **A1** and **A2**, the theoretical bound $\tau > 0.54518$ quite often excludes common datasets because of its relative high lower $\tau$ bound value compared to other copulas. Also, and the likelihood surface becomes unusually flat for

moderate–large $\theta$, undermining gradient-based refinement. these structural limitations necessitate bypassing both moment inversion and likelihood optimization, motivating our neural framework IGNIS.

The pathologies discussed above raise a crucial question: are these not indicators of fundamentally flawed models? We argue that this perspective is precisely what motivates our work. The A1 and A2 copulas offer unique theoretical advantages, such as capturing dual tail-dependence, but their "questionable properties", a flat likelihood surface and a restrictive theoretical range for Kendall's $\tau$ are the very barriers that make them unusable with classical methods. The goal of this paper is not to defend these models as universally optimal, but rather to introduce the first viable estimation framework that makes them accessible for practical application and empirical critique. By developing IGNIS, we provide the necessary tool for researchers to finally apply these models to real-world data and investigate the practical implications of their unusual theoretical structures, a task that was previously computationally infeasible.

## 6 Methodology: IGNIS Network

Named after the Latin word for "fire," the IGNIS Network is a unified neural estimator for four Archimedean copula families (Gumbel, Joe, A1, A2), each with the same parameter domain $\theta \geq 1$.

**Reproducibility:** All experiments use a fixed seed (123) applied globally across Python's `random` module, NumPy, TensorFlow, and PyTorch to ensure full computational reproducibility. Code runs on Python 3.11 with TensorFlow 2.19, SciPy 1.15.3, and scikit-learn 1.6.1.

**Input Representation:** Each example is a 9-D vector $x = [\mathbf{f}; \mathbf{c}]$, where

**1.** $\mathbf{f} \in \mathbb{R}^5$ consists of five dependency measures: empirical Kendall's $\tau$, Spearman's $\rho$, upper tail-dependence at the 0.95 quantile ($\lambda_{\mathrm{upper}}$), lower tail-dependence at the 0.05 quantile ($\lambda_{\mathrm{lower}}$), and the Pearson correlation coefficient ($r$).

**2.** $\mathbf{c} \in \{0,1\}^4$ is a one-hot encoded vector identifying the copula family.

**Network Architecture:** Let $\mathbf{x} \in \mathbb{R}^9$. We apply:

$$h_1 = \mathrm{ReLU}(W_1 x + b_1), \quad (128)$$
$$h_2 = \mathrm{ReLU}(W_2 h_1 + b_2), \quad (128)$$
$$h_3 = \mathrm{ReLU}(W_3 h_2 + b_3), \quad (64)$$
$$\theta_{\mathrm{raw}} = W_4 h_3 + b_4 \in \mathbb{R}.$$

A `softplus` activation plus 1 enforces $\hat{\theta} \geq 1$:

$$\hat{\theta} = \mathrm{softplus}(\theta_{\mathrm{raw}}) + 1.$$

Figure 2 illustrates this flow.

**Training Data Generation:** For each family, we sample 500 $\theta$ values uniformly from the range $[1, 20]$. For each $\theta$, we simulate $n = 5{,}000$ pairs $(U, V)$ using Algorithm 1(Section 4.3), compute the five summary features for the vector $f$, and concatenate the corresponding one-hot vector $c$. This process yields a total of $500 \times 4 = 2000$ training examples.

**Feature Scaling:** We standardize all 9-D inputs using scikit-learn's `StandardScaler`. The scaler is fitted only on the training data split and then applied to transform both the validation and test sets. We note that while standardizing the one-hot encoded portion of the input vector is not strictly necessary, we do so here for pipeline uniformity; this linear transformation has no adverse effect on the model's performance.

**Hyperparameters:** In Table 2 we see that training uses MSE loss with Adam (Kingma & Ba, 2015) $(5 \times 10^{-4})$, batch size 32, max 200 epochs, early stopping (patience 20 on 20% validation).

**Uncertainty Quantification in Simulations:** To rigorously evaluate the stability of the IGNIS estimator in our simulation studies, we employed a replication-based approach. For each copula family and each true $\theta$ value, the entire data generation and estimation process was repeated 100 times. This produced a

Table 2: Key Hyperparameters

| Hyperparameter | Value |
|---|---|
| Batch size | 32 |
| Learning rate | $5 \times 10^{-4}$ |
| Optimizer | Adam |
| Max epochs | 200 |
| Early-stop patience | 20 |
| Train/val split | 80/20 |
| Simulation replications | 100 |
| Bootstrap replicates (real data) | 100 |

distribution of 100 independent point estimates ($\hat{\theta}$). The standard deviation of this distribution serves as a direct and robust measure of the estimator's precision.

**Implementation Details:** IGNIS is implemented in TensorFlow/Keras with He-uniform initialization for all Dense layers. All training was performed on an NVIDIA GeForce RTX 4060 Laptop GPU.

**Theoretical Soundness:** One-hot encoding ensures family identifiability. Under regularity conditions (Appendix C), $\hat{\theta} \xrightarrow{p} \theta$. The softplus+1 transform guarantees $\hat{\theta} \in [1, \infty)$.

Figure 2 illustrates the IGNIS architecture. A 9-D input vector (five dependency measures + 4-D one-hot family ID) is processed by three fully connected layers (128–128–64 ReLU, He-uniform), and a final softplus+1 activation guarantees $\hat{\theta} \geq 1$.

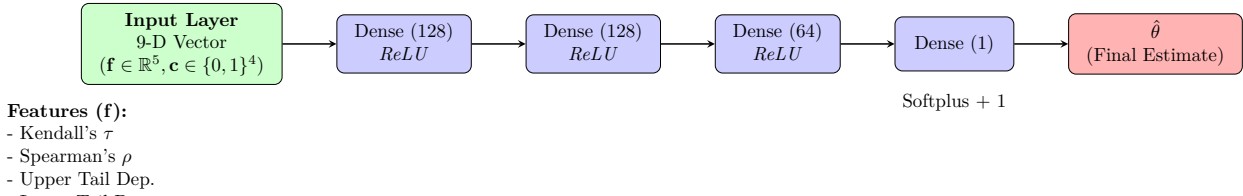

**Features (f):**
- Kendall's $\tau$
- Spearman's $\rho$
- Upper Tail Dep.
- Lower Tail Dep.
- Pearson $r$

Figure 2: The updated IGNIS Architecture. A 9-D input vector (five dependency measures, **f**, and a 4-D one-hot family identifier, **c**) is processed by three ReLU-activated hidden layers. A final dense layer followed by a Softplus+1 activation enforces the constraint $\hat{\theta} \geq 1$.

## 7 Simulation Studies for IGNIS

The same simulation setup described in Section 6 is followed here for $\theta = \{2.0, 5.0, 10.0, 15.0, 20.0\}$.

Table 3 show performance of the IGNIS network on simulated data.

**Key observations**: Table 3 provides a comprehensive performance evaluation of the IGNIS estimator across a wide spectrum of dependence levels, from weak ($\theta = 2.0$) to extreme ($\theta = 20.0$). For a broad and practical operational range (approximately $\theta \in [2, 15]$), the estimator demonstrates excellent properties. The **Bias** is consistently low, and the Root Mean Squared Error (RMSE) is driven almost entirely by the estimator's low variance (Std. Dev.), indicating both high accuracy and precision.

At the extreme end of the tested range ($\theta = 20.0$), which represents a region of intense dependence where classical methods are computationally infeasible, IGNIS maintains high precision for all families but exhibits a notable underestimation bias for the most challenging A1 and A2 copulas. For the A1 family, this bias ($-1.52$) becomes the dominant component of the RMSE. This detailed analysis validates IGNIS as a robust and reliable estimator for a wide array of practical scenarios while also rigorously characterizing its operational boundaries. This provides a clear and honest performance benchmark for the first viable estimation tool for

Table 3: IGNIS Network Performance Metrics from Simulation Study. Each metric is calculated based on test datasets of size $n = 5,000$.

| Copula | True $\theta$ | Est. $\theta$ | Bias | Std. Dev. (100 Runs) | RMSE |
|--------|------|------|------|------|------|
| | | | $\theta = 2.0$ | | |
| Gumbel | 2 | 2.12 | 0.12 | 0.05 | 0.13 |
| Joe | 2 | 1.96 | $-0.04$ | 0.06 | 0.07 |
| A1 | 2 | 1.97 | $-0.03$ | 0.10 | 0.11 |
| A2 | 2 | 1.91 | $-0.09$ | 0.08 | 0.13 |
| | | | $\theta = 5.0$ | | |
| Gumbel | 5 | 5.15 | 0.15 | 0.19 | 0.24 |
| Joe | 5 | 5.05 | 0.05 | 0.10 | 0.11 |
| A1 | 5 | 5.10 | 0.10 | 0.12 | 0.16 |
| A2 | 5 | 4.97 | $-0.03$ | 0.13 | 0.13 |
| | | | $\theta = 10.0$ | | |
| Gumbel | 10 | 10.17 | 0.17 | 0.17 | 0.24 |
| Joe | 10 | 9.92 | $-0.08$ | 0.18 | 0.20 |
| A1 | 10 | 10.10 | 0.10 | 0.26 | 0.28 |
| A2 | 10 | 10.05 | 0.05 | 0.21 | 0.22 |
| | | | $\theta = 15.0$ | | |
| Gumbel | 15 | 15.16 | 0.16 | 0.27 | 0.31 |
| Joe | 15 | 14.77 | $-0.23$ | 0.27 | 0.35 |
| A1 | 15 | 15.34 | 0.34 | 0.22 | 0.40 |
| A2 | 15 | 15.58 | 0.58 | 0.21 | 0.62 |
| | | | $\theta = 20.0$ | | |
| Gumbel | 20 | 19.51 | $-0.49$ | 0.27 | 0.56 |
| Joe | 20 | 19.90 | $-0.10$ | 0.32 | 0.34 |
| A1 | 20 | 18.48 | $-1.52$ | 0.19 | 1.53 |
| A2 | 20 | 18.89 | $-1.11$ | 0.24 | 1.13 |

these complex families. To provide further objective validation of our estimator's quality, we present a out of sample log-likelihood comparison between IGNIS and MoM estimates in Table 4.

### 7.1 Out-of-sample log-likelihood comparison (IGNIS vs MoM)

We compare the *out-of-sample* log-likelihood achieved on held-out data when plugging in point estimates from **IGNIS** and from the **Method of Moments (MoM)**. For each setting we simulate $n = 5,000$ pairs using Algorithm 1, evaluate both estimators' fixed $\hat{\theta}$ on the same held-out sample (no re-optimization), and average over 100 replications.

To ensure numerical stability for A1/A2, we evaluate the copula density using the standard Archimedean identity (via the inverse-function theorem; see, e.g., Nelsen (2006); Joe (2014)):

$$c(u, v) = \psi''(w)\, \phi'(u)\, \phi'(v), \qquad w = \phi^{-1}\big(\phi(u) + \phi(v)\big),$$

with

$$\psi''(w) = -\frac{\phi''(w)}{\{\phi'(w)\}^3}.$$

Because $\phi'(t) < 0$ for Archimedean generators, we implement this as $(-\phi'(u))(-\phi'(v))$ so all multiplicative factors inside the log are positive. We then work entirely in log-space, clip pseudo-observations to $(\varepsilon, 1 - \varepsilon)$

(slightly stronger $\varepsilon$ for A1), use analytic safe inverses with a floored discriminant, and floor positive factors entering logs at $10^{-300}$ (numerical recommendations in Hofert et al. (2013)).

Table 4: Averaged out-of-sample log-likelihood on held-out samples for A1 and A2. Each entry averages 100 replications with $n = 5{,}000$ pairs. $\Delta$ is IGNIS $-$ MoM.

| True $\theta$ | Copula | Mean LL (MoM) | Mean LL (IGNIS) | $\Delta$ (IGNIS-MoM) |
|---|---|---|---|---|
| 2.0 | A1 | 5128.00 | 5128.92 | 0.92 |
| 2.0 | A2 | 5496.19 | 5498.02 | 1.83 |
| 5.0 | A1 | 9534.16 | 9532.97 | -1.19 |
| 5.0 | A2 | 10015.15 | 10015.58 | 0.44 |
| 10.0 | A1 | 12976.38 | 12976.13 | -0.25 |
| 10.0 | A2 | 13480.20 | 13492.20 | 12.00 |

**Takeaway.** IGNIS and MoM attain virtually identical out-of-sample likelihoods whenever MoM is applicable. Differences are numerically negligible (e.g., for A2 at $\theta = 10$, $\Delta = 12$ over 5,000 pairs $\approx 0.0024$ nats per sample). Thus, IGNIS matches MoM in its valid regime while remaining usable when MoM is not when empirical Kendall's $\tau < 0.54518$ for A1/A2.

## 8 Real-World Applications

We validate IGNIS using two distinct domains where copulas are widely applied: financial markets (AAPL–MSFT stock returns) and public health (CDC Diabetes Dataset). These applications demonstrate the network's versatility across data types. For clarity, we emphasize this is an estimation methodology demonstration, not a copula selection analysis.

The IGNIS network estimates $\theta$ through the following standardized workflow:

**1. Data Preprocessing:**
*Financial Data*: Attain stationarity via log-returns:

$$r_t = \log\big(P_t / P_{t-1}\big),$$

where $P_t$ are adjusted closing prices.
*Healthcare Data*: We use original variables (GenHlth, PhysHlth) without differencing.
For both domains, transform marginals to pseudo-observations via rank-based PIT:

$$u_i = \frac{\operatorname{rank}(x_i)}{n+1}, \quad v_i = \frac{\operatorname{rank}(y_i)}{n+1},$$

yielding $\{(u_i, v_i)\}_{i=1}^{n} \in [0,1]^2$ with approximately uniform margins. We divide by n+1 following standard practice for the empirical probability integral transform; this ensures the pseudo-observations lie strictly within the open unit interval (0,1), avoiding potential numerical issues with copula functions at the boundaries.

**2. Feature Extraction:**
From the paired pseudo-observations, we compute five dependence measures: (1) Empirical Kendall's $\tau$, (2) Spearman's $\rho$, (3) upper tail-dependence $\lambda_{upper} = \frac{1}{n}\sum_{i=1}^{n} \mathbf{1}\{u_i > 0.95, v_i > 0.95\}$, (4) lower tail-dependence $\lambda_{lower} = \frac{1}{n}\sum_{i=1}^{n} \mathbf{1}\{u_i < 0.05, v_i < 0.05\}$, and (5) the Pearson correlation coefficient. These form the feature vector $\mathbf{f} \in \mathbb{R}^5$.

**3. Input Construction:**
The feature vector $\mathbf{f}$ is concatenated with a one-hot encoded copula identifier $\mathbf{c} \in \{0,1\}^4$ for the families Gumbel, Joe, A1, and A2. This creates the final 9-dimensional input vector $\mathbf{x} = [\mathbf{f}; \mathbf{c}]$. This vector is then standardized using the STANDARDSCALER that was fitted on the simulated training data.

**4. Theta Estimation:**
The network architecture consists of three hidden layers with 128, 128, and 64 ReLU-activated units, each

initialized using the He initialization scheme. The final layer applies a softplus activation followed by a unit shift to guarantee that $\hat{\theta} \geq 1$. We train the IGNIS network using the Adam optimizer with a learning rate of $5 \times 10^{-4}$ and a mean-squared error loss function for 200 epochs. During training, 20% of the data are held out for validation, and early stopping with a patience of 20 epochs is employed to prevent overfitting.

**5. Uncertainty Quantification:** To quantify the uncertainty of our estimates on the real-world datasets, we perform a bootstrap procedure. For each dataset, we resample the pseudo-observations with replacement $B = 100$ times. For each bootstrap resample, we recompute the five summary features and obtain a corresponding estimate $\hat{\theta}^{(b)}$. The bootstrap standard error of $\hat{\theta}$ is then calculated as the sample standard deviation of these bootstrap estimates:

$$\widehat{\mathrm{SE}}(\hat{\theta}) = \mathrm{std}\big(\{\hat{\theta}^{(b)}\}_{b=1}^{B}\big).$$

Results for both applications are presented in Tables 5 and 6, following identical estimation protocols for cross-domain comparability.

## 8.1 Dataset 1: AAPL-MSFT Returns Dataset

**Source and Period:** The dataset (Aroussi, 2024) comprises daily adjusted closing prices for two stocks, AAPL and MSFT, obtained from `yfinance` library in Python. Data were collected for the period from January 1, 2020 to December 31, 2023.

**Variables:** The primary variable of interest is the adjusted closing price for each ticker. This column (labeled either as `Adj Close` or `Close`) reflects the price after accounting for corporate actions such as dividends and stock splits.

**Derived Measures:** From the raw price data, daily log returns are computed. These log returns serve as a proxy for the instantaneous rate of return and are stationary.

### 8.1.1 Estimation Results

Table 5 summarizes the parameter estimation.

Table 5: Estimated $\theta$ Values and Bootstrap Standard Errors from Financial Data

| Copula | Estimated $\theta$ | Bootstrap SE($\theta$) |
|--------|--------------------|------------------------|
| Gumbel | 2.6755 | 0.2107 |
| Joe | 3.4592 | 0.3273 |
| A1 | 1.3332 | 0.0756 |
| A2 | 1.2140 | 0.0776 |

## 8.2 Dataset 2: CDC Diabetes Dataset

**Source:** We programmatically retrieved the CDC Diabetes Health Indicators dataset (UCI ML Repository ID 891) using the `ucimlrepo` Python package (Centers for Disease Control and Prevention, 2023). The full dataset contains 253,680 respondents and 21 original features; for our analysis we pulled only the two raw columns `GenHlth` and `PhysHlth`.

**Variables:** From these two columns we constructed empirical pseudo-observations via the probability integral transform (PIT), i.e.

$$u_i = \frac{\mathrm{rank}(GenHlth_i)}{n+1}, \quad v_i = \frac{\mathrm{rank}(PhysHlth_i)}{n+1},$$

where $n = 253{,}680$. These appear in our pipeline as:

1. `GenHlth_pu`: $u_i$, the pseudo-value for general health

2. PhysHlth_pu: $v_i$, the pseudo-value for physical health

### 8.2.1 Estimation Results

Table 6 summarizes the parameter estimation.

Table 6: Estimated $\theta$ Values and Bootstrap Standard Errors from CDC Diabetes Data

| Copula | Estimated $\theta$ | Bootstrap SE($\theta$) |
|--------|--------------------|------------------------|
| Gumbel | 1.4393 | 0.0031 |
| Joe | 2.6060 | 0.0064 |
| A1 | 1.3941 | 0.0050 |
| A2 | 1.2393 | 0.0024 |

### 8.3 Discussion of Application Results

The results from the financial and public health applications are presented in Tables 4 and 5, respectively. For both datasets, the IGNIS network produces stable parameter estimates. The bootstrap standard errors, which quantify the estimator's variance, are consistently small. For instance, in the high-sample CDC dataset, the SE values are exceptionally low (e.g., 0.0024 for the A2 copula), indicating that the learned estimation function is robust to small perturbations in the input data and yields consistent results across bootstrap resamples.

## 9 Conclusion and Future Work

In this paper, we confronted the critical failure of classical estimation methods when applied to an important class of Archimedean copulas with pathological likelihoods. We demonstrated that numerical instabilities, high Kendall's $\tau$ values and vanishing gradients make traditional inference via Maximum Likelihood or the Method of Moments inconsistent and computationally infeasible. To solve this, we introduced the **IGNIS Network**, a deep learning framework that provides robust, constraint-aware parameter estimates by learning a direct mapping from data-driven statistics. By leveraging a multi-layer architecture and a theory-guided `softplus+1` output layer, IGNIS delivers accurate and stable estimates for multiple copula families, succeeding precisely where classical methods fail. Beyond methodological innovation, IGNIS has broad **practical implications**: in extreme-value analysis, A1/A2's dual tail-dependence structure enables risk analysts to reliably model joint tail events (e.g., market crashes or insurance claims during natural disasters), improving capital allocation and hedging strategies. In anomaly detection for industrial IoT networks, it identifies coordinated failure patterns where sensors exhibit asymmetric tail dependencies. In healthcare, it models comorbid extreme health episodes where patients experience simultaneous deterioration of multiple health indicators. By solving the parameter estimation challenge for these advanced copulas, IGNIS unlocks their potential for **real-time risk assessment** and **multivariate anomaly detection systems**.

Despite these strengths, IGNIS has several limitations. First, our evaluation has been restricted to the class of bivariate Archimedean families where theta greater than or equal to one. Integrating commonly used generators with different parameter domains is a key direction for future work. For instance, the Clayton family, with its full parameter domain of theta in $[-1, \infty) \setminus \{0\}$, could be incorporated by modifying the output layer (e.g., using a `softplus(z) - 1` activation). Similarly, the Frank copula (theta in $\mathbb{R} \setminus \{0\}$) would require its own architectural adaptation, such as using a scaled `tanh` activation to map to its broad real-valued domain. Second, the current architecture handles only two-dimensional dependencies, so extending to multivariate or nested copulas will require permutation-invariant or graph-based neural designs. Third, reliance on a fixed set of four summary statistics may limit performance in small-sample or heavy-tailed scenarios, suggesting that adaptive or richer feature representations could enhance robustness. Finally, IGNIS assumes a known family identifier via one-hot encoding, leaving fully automated copula selection as an open challenge.

Looking ahead, we see several promising directions for future work. Incorporating Clayton, Frank, and other Archimedean generators will broaden IGNIS's applicability. High-dimensional extensions can be pursued by designing architectures, such as DeepSets or attention-based graphs, that respect permutation symmetry in multivariate dependence. To capture dynamic relationships, we plan to integrate recurrent or temporal-attention modules that adapt to time-varying copulas. A comprehensive search for the optimal network architecture, while beyond the scope of this paper, could also yield performance improvements. Additionally, an empirical ablation study comparing the performance of our unified model against separately trained networks could offer further insights into architectural choices. We can use alternative features (e.g., Blomqvist's $\beta$, Gini's $\gamma$) in the future. Joint inference of copula family and parameter via mixture-of-experts or multi-task learning would eliminate the need for a priori family tagging. Also, we plan to conduct a rigorous comparative performance study between the IGNIS framework and global optimization methods, such as Particle Swarm Optimization (PSO) and Genetic Algorithms (GA). On the uncertainty front, embedding Bayesian neural networks or deep ensembles can provide principled credible intervals for $\hat{\theta}$. Again, exploring alternative summary features, such as higher-order tail-dependence coefficients or distance-based metrics, may further improve estimation under challenging data regimes. Furthermore, exploring end-to-end architectures that learn feature representations directly from raw pseudo-observations, rather than relying on a fixed set of summary statistics, presents another promising avenue for future research. Together, these extensions will help establish IGNIS as a comprehensive, data-driven toolkit for dependence modeling across diverse applications.

### Broader Impact Statement

The primary motivation for this work is to provide a positive methodological contribution to the statistics and machine learning communities. By creating a reliable estimator, IGNIS, for complex copula models like A1 and A2, we enable researchers and practitioners to use more appropriate and flexible models in fields like financial risk management and public health analytics, which was previously computationally infeasible.

The main broader consideration is standard model risk. As with any powerful statistical tool, there is a potential for misuse if it is applied without a proper understanding of its context. For example, a user could generate a precise parameter estimate for a copula family that is fundamentally a poor fit for their data, leading to a false sense of security. To mitigate this, we emphasize that IGNIS is a tool for parameter estimation, not model selection, and must be used as part of a larger workflow that includes rigorous goodness-of-fit testing.

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

# A Appendix: Full Derivation of Kendall's $\tau$ for A1 and A2 Copulas

In this appendix, we derive explicit analytical expressions for Kendall's $\tau$ for the novel Archimedean copulas A1 and A2. These derivations form the theoretical basis for the Method-of-Moments estimation of the copula parameter $\theta$.

## A.1 Derivation for the A1 Copula

For a general Archimedean copula with generator $\phi(t)$, Kendall's $\tau$ is given by

$$\tau = 1 + 4 \int_0^1 \frac{\phi(t)}{\phi'(t)} dt.$$

For the A1 copula the generator is

$$\phi_{A1}(t;\theta) = (t^{1/\theta} + t^{-1/\theta} - 2)^\theta, \quad \theta \geq 1.$$

**Step 1: Differentiation of $\phi_{A1}(t;\theta)$.** The derivative of the generator with respect to $t$ is found using the chain rule:

$$\phi'_{A1}(t;\theta) = \theta(t^{1/\theta} + t^{-1/\theta} - 2)^{\theta-1}\left[\frac{1}{\theta}t^{1/\theta-1} - \frac{1}{\theta}t^{-1/\theta-1}\right].$$

Cancelling the factor of $\theta$, we get:

$$\phi'_{A1}(t;\theta) = (t^{1/\theta} + t^{-1/\theta} - 2)^{\theta-1}[t^{1/\theta-1} - t^{-1/\theta-1}].$$

**Step 2: Form the Ratio $\phi_{A1}/\phi'_{A1}$.** Taking the ratio of the generator and its derivative simplifies to:

$$\frac{\phi_{A1}(t;\theta)}{\phi'_{A1}(t;\theta)} = \frac{(t^{1/\theta} + t^{-1/\theta} - 2)^{\theta}}{(t^{1/\theta} + t^{-1/\theta} - 2)^{\theta-1}[t^{1/\theta-1} - t^{-1/\theta-1}]}$$
$$= \frac{t^{1/\theta} + t^{-1/\theta} - 2}{t^{1/\theta-1} - t^{-1/\theta-1}}.$$

Further algebraic simplification shows that this expression is equivalent to:

$$\frac{\phi_{A1}(t;\theta)}{\phi'_{A1}(t;\theta)} = \frac{t(t^{1/\theta} - 1)}{1 + t^{1/\theta}}.$$

**Step 3: Change of Variables.** To evaluate the integral, we set $u = t^{1/\theta}$, which implies $t = u^{\theta}$ and $dt = \theta u^{\theta-1}du$. Substituting these into the integral from the corrected ratio in Step 2 gives:

$$I(\theta) = \int_0^1 \frac{\phi_{A1}(t;\theta)}{\phi'_{A1}(t;\theta)}dt = \int_0^1 \frac{t(t^{1/\theta} - 1)}{1 + t^{1/\theta}}dt$$
$$= \int_0^1 \frac{u^{\theta}(u - 1)}{1 + u}(\theta u^{\theta-1})du$$
$$= \theta \int_0^1 \frac{u^{2\theta-1}(u - 1)}{1 + u}du$$
$$= \theta \int_0^1 \frac{u^{2\theta} - u^{2\theta-1}}{1 + u}du.$$

**Step 4: Evaluate the Integral.** The integral can be solved using a standard identity for the digamma function, $\psi(\cdot)$, where:

$$\int_0^1 \frac{x^a - x^b}{1 + x}dx = \frac{1}{2}\left[\psi\left(\frac{a+2}{2}\right) - \psi\left(\frac{a+1}{2}\right) - \psi\left(\frac{b+2}{2}\right) + \psi\left(\frac{b+1}{2}\right)\right].$$

Setting $a = 2\theta$ and $b = 2\theta - 1$, the integral part becomes:

$$\int_0^1 \frac{u^{2\theta} - u^{2\theta-1}}{1 + u}du = \frac{1}{2}\left[\psi(\theta + 1) - \psi\left(\theta + \frac{1}{2}\right) - \psi\left(\theta + \frac{1}{2}\right) + \psi(\theta)\right]$$
$$= \frac{1}{2}\left[\psi(\theta + 1) + \psi(\theta) - 2\psi\left(\theta + \frac{1}{2}\right)\right].$$

Using the recurrence relation $\psi(\theta + 1) = \psi(\theta) + 1/\theta$, this simplifies to:

$$\frac{1}{2}\left[(\psi(\theta) + \frac{1}{\theta}) + \psi(\theta) - 2\psi\left(\theta + \frac{1}{2}\right)\right] = \psi(\theta) - \psi\left(\theta + \frac{1}{2}\right) + \frac{1}{2\theta}.$$

Finally, we multiply by the leading factor of $\theta$ from Step 3:

$$I(\theta) = \theta\left[\psi(\theta) - \psi\left(\theta + \frac{1}{2}\right) + \frac{1}{2\theta}\right] = \theta\left[\psi(\theta) - \psi\left(\theta + \frac{1}{2}\right)\right] + \frac{1}{2}.$$

**Step 5: Final Expression for A1.** Substituting the correct integral value back into the formula for Kendall's $\tau$, we obtain the final expression:

$$\tau_{A1} = 1 + 4I(\theta) = 1 + 4\left(\theta\left[\psi(\theta) - \psi\left(\theta + \frac{1}{2}\right)\right] + \frac{1}{2}\right).$$

This implies:

$$\boxed{\tau_{A1} = 3 + 4\theta\left[\psi(\theta) - \psi\left(\theta + \frac{1}{2}\right)\right].}$$

### A.2 Derivation for the A2 Copula

For the A2 copula, the generator is defined as

$$\phi_{A2}(t;\theta) = \left(\frac{1}{t}(1-t)^2\right)^{\theta}, \quad \theta \geq 1.$$

Following a similar differentiation process (details omitted here), one obtains

$$\frac{\phi_{A2}(t;\theta)}{\phi'_{A2}(t;\theta)} = \frac{t(t-1)}{\theta(t+1)}.$$

Thus, Kendall's $\tau$ is given by

$$\tau_{A2} = 1 + 4\int_0^1 \frac{\phi_{A2}(t;\theta)}{\phi'_{A2}(t;\theta)}\,dt = 1 + \frac{4}{\theta}\int_0^1 \frac{t(t-1)}{t+1}\,dt.$$

**Step 1: Evaluate the Integral** Define

$$J = \int_0^1 \frac{t(t-1)}{t+1}\,dt.$$

Since

$$t(t-1) = t^2 - t,$$

we perform polynomial division of $t^2 - t$ by $t + 1$. Dividing, we obtain

$$\frac{t^2 - t}{t+1} = t - 2 + \frac{2}{t+1}.$$

Thus,

$$J = \int_0^1 \left(t - 2 + \frac{2}{t+1}\right)dt.$$

**Step 2: Integrate Term-by-Term** We compute each integral:

$$\int_0^1 t\,dt = \frac{t^2}{2}\bigg|_0^1 = \frac{1}{2},$$

$$\int_0^1 dt = 1,$$

$$\int_0^1 \frac{1}{t+1}\,dt = \ln|t+1|\big|_0^1 = \ln 2.$$

Hence,

$$J = \frac{1}{2} - 2\cdot 1 + 2\ln 2 = \frac{1}{2} - 2 + 2\ln 2 = -\frac{3}{2} + 2\ln 2.$$

**Step 3: Final Expression for A2** Substituting back into the expression for $\tau_{A2}$, we have

$$\boxed{\tau_{A2} = 1 - \frac{6 - 8\ln 2}{\theta}.}$$

### A.3 Strict monotonicity of $\tau_{A1}(\theta)$

$$\tau_{A1}(\theta) = 3 + 4\theta[\psi(\theta) - \psi(\theta + \frac{1}{2})], \quad \theta \geq 1.$$

**Claim.** $\tau_{A1}$ is strictly increasing on $[1, \infty)$; moreover

$$\tau_{A1}(1) = 8\ln 2 - 5 \approx 0.54518, \quad \lim_{\theta \to \infty} \tau_{A1}(\theta) = 1.$$

**Proof.** Set

$$f(\theta) := \psi(\theta) - \psi\left(\theta + \frac{1}{2}\right).$$

A standard integral representation of the digamma function yields, for $x > 0$ and $a > 0$,

$$\psi(x) - \psi(x + a) = -\int_0^\infty \frac{1 - e^{-at}}{1 - e^{-t}} e^{-xt} dt.$$

With $a = \frac{1}{2}$ and $x = \theta$ we get

$$f(\theta) = -\int_0^\infty H(t)e^{-\theta t} dt, \quad H(t) := \frac{1 - e^{-t/2}}{1 - e^{-t}}, \quad t > 0.$$

Hence,

$$\tau_{A1}(\theta) = 3 - 4\theta \int_0^\infty H(t)e^{-\theta t} dt. \tag{A}$$

**(i) Value at $\theta = 1$.** Using $\psi(1) = -\gamma$ and $\psi\left(\frac{3}{2}\right) = \psi\left(\frac{1}{2}\right) + 2 = -\gamma - 2\ln 2 + 2$,

$$\tau_{A1}(1) = 3 + 4\left[\psi(1) - \psi\left(\frac{3}{2}\right)\right] = 3 + 4(2\ln 2 - 2) = 8\ln 2 - 5.$$

**(ii) Limit as $\theta \to \infty$.** Near $t = 0$, $H(t)$ is continuous with $H(0) := \lim_{t\downarrow 0} \frac{1-e^{-t/2}}{1-e^{-t}} = \frac{1}{2}$. Since $\theta e^{-\theta t}$ is an approximate identity on $[0, \infty)$,

$$\theta \int_0^\infty H(t)e^{-\theta t} dt \longrightarrow H(0) = \frac{1}{2}.$$

Taking this limit in (A) gives

$$\lim_{\theta \to \infty} \tau_{A1}(\theta) = 3 - 4 \cdot \frac{1}{2} = 1.$$

**(iii) Strict monotonicity on $[1, \infty)$.** Differentiate (A):

$$\tau'_{A1}(\theta) = 4(f(\theta) + \theta f'(\theta)) = 4\int_0^\infty (\theta t - 1)H(t)e^{-\theta t} dt. \tag{B}$$

We now show the right-hand side is strictly positive for every $\theta > 0$ (hence for $\theta \geq 1$).

First, observe that $H$ is strictly increasing on $(0, \infty)$. Indeed,

$$
\begin{aligned}
H'(t) &= \frac{\frac{1}{2}e^{-t/2}(1-e^{-t}) - e^{-t}(1-e^{-t/2})}{(1-e^{-t})^2} \\
&= \frac{\frac{1}{2}e^{-t/2}(1-e^{-t/2})^2}{(1-e^{-t})^2} \\
&= \frac{1}{2}\frac{e^{-t/2}}{(1+e^{-t/2})^2} > 0.
\end{aligned}
$$

Next, rewrite (B) by subtracting a zero term and integrating by parts in a monotone way. Since

$$
\int_0^\infty (\theta t - 1)e^{-\theta t}dt = 0,
$$

$$
\int_0^\infty (\theta t - 1)H(t)e^{-\theta t}dt = \int_0^\infty (\theta t - 1)[H(t) - H(0)]e^{-\theta t}dt.
$$

Write $H(t) - H(0) = \int_0^t H'(s)ds$, interchange integrals, and evaluate the inner integral:

$$
\int_s^\infty (\theta t - 1)e^{-\theta t}dt = [-te^{-\theta t}]_{t=s}^\infty = se^{-\theta s}.
$$

Therefore,

$$
\int_0^\infty (\theta t - 1)H(t)e^{-\theta t}dt = \int_0^\infty se^{-\theta s}H'(s)ds.
$$

Since $s > 0$, $e^{-\theta s} > 0$, and $H'(s) > 0$ for all $s > 0$, the integrand is strictly positive on $(0, \infty)$, hence the integral is strictly positive. Combining with (B),

$$
\tau_{A1}'(\theta) = 4\int_0^\infty se^{-\theta s}H'(s)ds > 0 \quad (\theta > 0).
$$

In particular, $\tau_{A1}$ is strictly increasing on $[1, \infty)$.

This completes the proof. $\square$

**Remark (explicit positive form).** Using the closed form $H'(t) = \frac{1}{2}e^{-t/2}/(1+e^{-t/2})^2$, the derivative can be written as

$$
\tau_{A1}'(\theta) = 2\int_0^\infty \frac{se^{-(\theta + \frac{1}{2})s}}{(1+e^{-s/2})^2}ds > 0,
$$

making strict positivity immediate.

## B   Appendix: Identifiability Proofs for A1 and A2 Copulas

### B.1   A1 Copula Identifiability

For the A1 family, Kendall's $\tau$ has the closed form

$$
\tau_{A1}(\theta) = 3 + 4\theta\left[\psi(\theta) - \psi\left(\theta + \tfrac{1}{2}\right)\right], \qquad \theta \geq 1,
$$

and we showed in Appendix A.3 that $\tau_{A1}(\theta)$ is *strictly increasing* on $[1, \infty)$. Hence if $\theta_1 \neq \theta_2$ then $\tau_{A1}(\theta_1) \neq \tau_{A1}(\theta_2)$, so the induced copulas $C(\cdot, \cdot; \theta_1)$ and $C(\cdot, \cdot; \theta_2)$ are distinct. Therefore, the parameter $\theta$ is identifiable in the A1 family.

### B.2 A2 Copula Identifiability

For the A2 generator:

$$\phi_{A2}(t;\theta) = \left(\frac{(1-t)^2}{t}\right)^{\theta}, \quad \theta \geq 1,$$

assume $\phi_{A2}(t;\theta_1) = \phi_{A2}(t;\theta_2)$ for all $t \in (0,1)$. Taking logarithms:

$$\theta_1 \ln\left(\frac{(1-t)^2}{t}\right) = \theta_2 \ln\left(\frac{(1-t)^2}{t}\right).$$

For $t \neq \frac{3-\sqrt{5}}{2}$ (where $\frac{(1-t)^2}{t} \neq 1$), $\ln\left(\frac{(1-t)^2}{t}\right) \neq 0$. Hence:

$$(\theta_1 - \theta_2) \ln\left(\frac{(1-t)^2}{t}\right) = 0 \implies \theta_1 = \theta_2,$$

for all non-degenerate $t$, proving injectivity.

**B**oth proofs rigorously establish that $\phi_{\theta_1} = \phi_{\theta_2} \implies \theta_1 = \theta_2$, ensuring parameter identifiability for A1 and A2 copulas.

## C Consistency proof for A1 and A2 copulas

**Regularity Conditions.** For every copula family in {Gumbel, Joe, A1, A2}, we assume:
**1. Identifiability:** The mapping $\theta \mapsto \mathbf{T}(\theta)$ is injective within each family. In other words, if $\phi_{\theta_1} = \phi_{\theta_2}$ then $\theta_1 = \theta_2$. (See (Nelsen, 2006) for the Gumbel and Joe copulas; for the A1/A2 families we have given the proof in Appendix B.)
**2.** The generator $\phi_\theta$ is continuously differentiable in $\theta$.
**3. Feature Continuity:** The vector of summary features

$$\mathbf{T}_n = (\tau_n, \rho_n, \lambda_{upper,n}, \lambda_{lower,n}, r_n)$$

is continuous in $\theta$. Moreover, a standard lemma (established via Donsker's theorem for copula processes) shows that the empirical features converge uniformly to their population counterparts over the compact set $\Theta$.

**Theorem 1** *Assume the regularity conditions above hold and further suppose that:*
**1. Universal Approximation:** *There exists a neural network (NN) architecture that is dense in the space $\mathscr{C}(\Theta)$ of continuous functions on $\Theta$; here, we assume that $\Theta$ and the feature space $\mathcal{T}$ are compact, as required by Hornik's theorem (Hornik, 1991).*
**2. Training Density:** *As the number of training samples $N_{train} \to \infty$, the training data become dense over $\Theta$.*
**3. Operational Regime:** *The number of real observations $n \to \infty$.*
*Then the IGNIS estimator satisfies*

$$\hat{\theta}_n \xrightarrow{p} \theta_0 \quad as \ n \to \infty.$$

**Proof.** The proof proceeds in five steps.

**Step 1: Uniform Feature Convergence.** By a standard lemma (which follows from Donsker's theorem (van der Vaart & Wellner, 1996) for copulas), the empirical summary features converge uniformly (in probability) to the population features:

$$\sup_{\theta \in \Theta} \|\mathbf{T}_n(\theta) - \mathbf{T}_\infty(\theta)\| \xrightarrow{p} 0.$$

**Step 2: Identifiability.** Define the mapping $g^*(\mathbf{T}, C)$ as the true (population) function that maps the summary features and the copula type $C$ to the parameter $\theta$, where $C$ denotes the copula family. Then, by the injectivity of $\theta \mapsto \mathbf{T}(\theta)$ within each copula family (see above), if

$$g^*(\mathbf{T}^{(1)}, C^{(1)}) = g^*(\mathbf{T}^{(2)}, C^{(2)}),$$

it follows that $(\theta^{(1)}, C^{(1)}) = (\theta^{(2)}, C^{(2)})$.

**Step 3: Universal Approximation.** By the universal approximation theorem (Hornik, 1991), for any $\epsilon > 0$ there exist network parameters $W$ such that

$$\sup_{(\mathbf{T}, C) \in \mathcal{T} \times \mathcal{C}} \left| f_{\mathrm{NN}}(\mathbf{T}, C; W) - g^*(\mathbf{T}, C) \right| < \epsilon,$$

where we assume that both $\Theta$ and the feature set $\mathcal{T}$ are compact.

**Step 4: Training Risk Convergence.** Let the mean squared error (MSE) loss be defined as

$$\frac{1}{N_{\mathrm{train}}} \sum_{i=1}^{N_{\mathrm{train}}} \left( f_{\mathrm{NN}}(\mathbf{T}_i, C_i; W) - \theta_i \right)^2.$$

By White's Theorem (White, 1989), as $N_{\mathrm{train}} \to \infty$ this training loss converges to zero.

**Step 5: Operational Consistency.** Define $f_{\mathrm{NN}}(\mathbf{T}_\infty, C)$ as the neural network applied to the population features. Then, by a standard decomposition,

$$\left\| f_{\mathrm{NN}}(\mathbf{T}_n, C) - \theta_0 \right\| \leq \underbrace{\left\| f_{\mathrm{NN}}(\mathbf{T}_n, C) - f_{\mathrm{NN}}(\mathbf{T}_\infty, C) \right\|}_{(a)}$$

$$+ \underbrace{\left\| f_{\mathrm{NN}}(\mathbf{T}_\infty, C) - \theta_0 \right\|}_{(b)}.$$

Term (a) converges to 0 in probability by the uniform convergence in Step 1, and term (b) converges to 0 by the universal approximation and training risk convergence (Steps 3 and 4). Therefore, by Slutsky's theorem (Slutsky, 1925),

$$\hat{\theta}_n = f_{\mathrm{NN}}(\mathbf{T}_n, C) \xrightarrow{p} \theta_0.$$

This completes the proof. ∎

**Practical Considerations**

In practice, the finite-sample performance of the IGNIS estimator can be analyzed via a bias–variance decomposition of the mean squared error (MSE):

$$\mathbb{E}\left[(\hat{\theta}_n - \theta_0)^2\right] \leq K_1 \, n^{-1} \; + \; K_2 \, N_{\mathrm{train}}^{-1} \; + \; K_3 \, \epsilon^2,$$

where $K_1 \, n^{-1}$ represents the estimation error due to finite sample size, $K_2 \, N_{\mathrm{train}}^{-1}$ accounts for the approximation error from limited training data, and $K_3 \, \epsilon^2$ reflects the error due to the network architecture approximation. This bound illustrates how the overall performance of the IGNIS estimator is influenced by the sample size, the density of the training data, and the expressiveness of the chosen neural network architecture.

## D  Pathological Properties of A1/A2 Copulas

**Asymptotic regimes.** In the analyses below we work in two distinct limits:

1. **Density-blowup (Barrier 1):** take $t \to 0^+$ with $\theta$ fixed, to capture the boundary singularity of $\phi''(t; \theta)$.

2. **Score- and Hessian-decay (Barriers 2 & 3):** take $\theta \to \infty$ with $t \in (0, 1)$ fixed, to derive the $O(\theta^{-8})$, $O(\theta^{-3})$, $O(\theta^{-9})$, and $O(\theta^{-4})$ decay rates.

### D.1  Derivative Analysis and Computational Complexity

### D.1.1  First and Second Derivatives of A1 Generator

For $\phi_{A1}(t;\theta) = \left(t^{1/\theta} + t^{-1/\theta} - 2\right)^{\theta}$, let $g(t) = t^{1/\theta} + t^{-1/\theta} - 2$.

The first derivative is:

$$\phi'_{A1}(t) = \theta g(t)^{\theta-1} g'(t)$$

where

$$g'(t) = \frac{1}{\theta} t^{1/\theta-1} - \frac{1}{\theta} t^{-1/\theta-1} = \frac{1}{\theta} t^{-1/\theta-1}\left(t^{2/\theta} - 1\right)$$

The second derivative is:

$$\phi''_{A1}(t) = \theta(\theta-1)g(t)^{\theta-2}[g'(t)]^2 + \theta g(t)^{\theta-1} g''(t)$$

where

$$g''(t) = \frac{1}{\theta}\left(\frac{1}{\theta} - 1\right) t^{1/\theta-2} + \frac{1}{\theta}\left(\frac{1}{\theta} + 1\right) t^{-1/\theta-2}$$

### D.1.2  First and Second Derivatives of A2 Generator

For $\phi_{A2}(t;\theta) = \left(\frac{1-t}{t}\right)^{\theta} (1-t)^{\theta}$, we rewrite as:

$$\phi_{A2}(t;\theta) = (1-t)^{2\theta} t^{-\theta}$$

The derivatives are:

$$\phi'_{A2}(t) = -\theta\,(1-t)^{2\theta-1}\,t^{-\theta-1}\,(1+t)$$
$$\phi''_{A2}(t) = \theta\,(1-t)^{2\theta-2}\,t^{-\theta-2}\left[(\theta+1) + 2(\theta-1)\,t + (\theta-1)\,t^2\right]$$

**Lemma 1 (A1 Score-Decay Rate)** *For the A1 generator*

$$\phi_{A1}(t;\theta) = \left(t^{1/\theta} + t^{-1/\theta} - 2\right)^{\theta},$$

*the per-observation score satisfies*

$$\partial_\theta \log c(u,v;\theta) = O\left(\theta^{-8}\right),$$

*and hence for $n$ i.i.d. pairs,*

$$\left|\partial_\theta \ell(\theta)\right| = \sum_{i=1}^{n} O\left(\theta^{-8}\right) = O\left(n\,\theta^{-8}\right).$$

**Proof.** Let $L = \ln t$. First expand

$$t^{1/\theta} = e^{L/\theta} = 1 + \frac{L}{\theta} + \frac{L^2}{2\theta^2} + \frac{L^3}{6\theta^3} + \frac{L^4}{24\theta^4} + O\left(\frac{1}{\theta^5}\right),$$
$$t^{-1/\theta} = 1 - \frac{L}{\theta} + \frac{L^2}{2\theta^2} - \frac{L^3}{6\theta^3} + \frac{L^4}{24\theta^4} + O\left(\frac{1}{\theta^5}\right).$$

Hence

$$g(t) = t^{1/\theta} + t^{-1/\theta} - 2 = \frac{L^2}{\theta^2} + \frac{L^4}{12\theta^4} + O\left(\frac{1}{\theta^6}\right).$$

Differentiate:

$$g'(t) = \frac{1}{\theta}\left(t^{1/\theta-1} - t^{-1/\theta-1}\right) = \frac{L^2}{\theta^2\,t} + O\left(\frac{1}{\theta^4}\right), \quad g''(t) = O\left(\frac{1}{\theta^2}\right).$$

Write

$$\phi'_{A1}(t) = \theta\, g^{\theta-1}\, g', \quad \phi''_{A1}(t) = \theta(\theta-1)\, g^{\theta-2}[g']^2 + \theta\, g^{\theta-1}\, g''.$$

Then

$$\ln \phi'_{A1}(t) = \ln\theta + (\theta-1)\ln g + \ln g',$$

$$\ln \phi''_{A1}(t) = \ln[\theta(\theta-1)] + (\theta-2)\ln g + 2\ln g' + \ln\!\Big(1 + \tfrac{g''}{(\theta-1)g'}\Big).$$

Differentiating in $\theta$ gives, after a lengthy but straightforward series-expansion in $1/\theta$:

$$\partial_\theta \ln \phi''_{A1}(t) = \sum_{k=1}^{8} \frac{A_k(t)}{\theta^k} + O\!\Big(\frac{1}{\theta^9}\Big), \quad \partial_\theta \ln \phi'_{A1}(t) = \sum_{k=1}^{8} \frac{B_k(t)}{\theta^k} + O\!\Big(\frac{1}{\theta^9}\Big).$$

A direct coefficient-comparison (matching powers of $1/\theta$) shows

$$A_1(t) - 2B_1(t) = 0,\ A_2(t) - 2B_2(t) = 0,\ \ldots,\ A_7(t) - 2B_7(t) = 0,$$

and the *first* nonzero difference is

$$A_8(t) - 2B_8(t) \;=\; O(1).$$

Hence for one pair

$$\partial_\theta \log c(u,v;\theta) = \partial_\theta \ln \phi''_{A1}(w) - 2\,\partial_\theta \ln \phi'_{A1}(u) = O\!\Big(\frac{1}{\theta^8}\Big),$$

and summing over $n$ gives the result. ∎

**Lemma 2 (A2 Score-Decay Rate)** *For the A2 generator*

$$\phi_{A2}(t;\theta) = (1-t)^{2\theta}\, t^{-\theta},$$

*one finds*

$$\partial_\theta \log c(u,v;\theta) = O\big(\theta^{-3}\big),$$

*and thus* $|\partial_\theta \ell(\theta)| = O(n\,\theta^{-3})$.

**Proof.** Write

$$\ln \phi'_{A2}(t) = \ln\theta + (2\theta-1)\ln(1-t) - (\theta+1)\ln t + \ln(1+t),$$

$$\ln \phi''_{A2}(t) = \ln[\theta(\theta-1)] + (2\theta-2)\ln(1-t) - (\theta+2)\ln t + \ln Q(t,\theta),$$

where $Q(t,\theta)$ is a polynomial of degree 2 in $t$. Differentiating and expanding in $1/\theta$ yields

$$\partial_\theta \ln \phi''_{A2}(t) - 2\,\partial_\theta \ln \phi'_{A2}(t) = \frac{C_1(t)}{\theta^2} + \frac{C_2(t)}{\theta^3} + O\!\Big(\tfrac{1}{\theta^4}\Big),$$

with the $1/\theta$ and $1/\theta^2$ terms canceling exactly. The first nonzero remainder is $O(1/\theta^3)$. Hence per-observation $\partial_\theta \log c = O(1/\theta^3)$, and summing $n$ copies gives $O(n\,\theta^{-3})$. ∎

**Lemma 3 (Hessian-Decay Rates)** *Under the same setup as Lemmas D.1 and D.2, the second derivative of the log-likelihood,*

$$\partial_\theta^2 \ell(\theta) = \sum_{i=1}^{n} \partial_\theta^2 \log c(u_i, v_i; \theta),$$

*satisfies*

$$|\partial_\theta^2 \ell(\theta)| = \begin{cases} O\big(n\,\theta^{-9}\big), & A1, \\ O\big(n\,\theta^{-4}\big), & A2. \end{cases}$$

**Proof.** We differentiate once more the cancellation expansions from Lemmas D.1 and D.2:

**1. A1 case** From Lemma 1 we had, per observation,

$$\partial_\theta \log c(u, v; \theta) = \sum_{k=8}^{\infty} \frac{C_k}{\theta^k}, \quad C_8 \neq 0.$$

Differentiating in $\theta$ gives

$$\partial_\theta^2 \log c(u, v; \theta) = \sum_{k=8}^{\infty} (-k) \frac{C_k}{\theta^{k+1}} = O\left(\frac{1}{\theta^9}\right).$$

Summing over $n$ pairs yields $O(n\,\theta^{-9})$.

**2. A2 case** From Lemma 2 we had, per observation,

$$\partial_\theta \log c(u, v; \theta) = \frac{D_3}{\theta^3} + O\left(\frac{1}{\theta^4}\right), \quad D_3 \neq 0.$$

Differentiating gives

$$\partial_\theta^2 \log c(u, v; \theta) = -3 \frac{D_3}{\theta^4} + O\left(\frac{1}{\theta^5}\right) = O\left(\frac{1}{\theta^4}\right).$$

Summing across $n$ observations yields $O(n\,\theta^{-4})$.

This completes the proof. ∎

### D.2 Proof of Numerical Instability (Barrier 1)

**Theorem 2 (Asymptotic Singularity Behavior)** *The second derivatives of the A1 and A2 generators exhibit severe asymptotic behavior near the boundary $t \to 0^+$:*

*1. For A1:*
$$\left|\phi_{A1}''(t)\right| \sim \mathcal{O}\left(t^{-3}\right).$$

*2. For A2:*
$$\left|\phi_{A2}''(t)\right| \sim \mathcal{O}\left(t^{-\theta-2}\right).$$

**Proof. Part 1: A1 Generator Singularity Analysis**

Recall
$$\phi_{A1}(t; \theta) = \left(t^{1/\theta} + t^{-1/\theta} - 2\right)^\theta, \quad g(t) = t^{1/\theta} + t^{-1/\theta} - 2.$$

We have
$$\phi_{A1}''(t) = \theta(\theta - 1)\, g(t)^{\theta-2} \left[g'(t)\right]^2 + \theta\, g(t)^{\theta-1}\, g''(t),$$

with
$$g'(t) = \frac{1}{\theta} t^{-1/\theta-1}\left(t^{2/\theta} - 1\right) \sim -\frac{1}{\theta} t^{-1/\theta-1}, \quad g''(t) \sim \frac{1}{\theta}\left(\frac{1}{\theta} + 1\right) t^{-1/\theta-2}, \quad g(t) \sim t^{-1/\theta}.$$

Hence as $t \to 0^+$:

$$\phi_{A1}''(t) \sim \theta(\theta - 1)\left(t^{-1/\theta}\right)^{\theta-2}\left(-\tfrac{1}{\theta} t^{-1/\theta-1}\right)^2 + \theta\left(t^{-1/\theta}\right)^{\theta-1}\left(\tfrac{1}{\theta}\left(\tfrac{1}{\theta} + 1\right) t^{-1/\theta-2}\right)$$

$$= \frac{\theta - 1}{\theta} t^{-3} + \left(\tfrac{1}{\theta} + 1\right) t^{-3} = 2 t^{-3} = \mathcal{O}(t^{-3}).$$

**Part 2: A2 Generator Singularity Analysis**

Since
$$\phi_{A2}(t; \theta) = (1 - t)^{2\theta}\, t^{-\theta},$$

one finds (see main text) that

$$\phi''_{A2}(t) = \theta\,(1-t)^{2\theta-2}\,t^{-\theta-2}\left[(\theta+1) + 2(\theta-1)t + (\theta-1)t^2\right].$$

As $t \to 0^+$, only the $(\theta+1)$–term survives:

$$\phi''_{A2}(t) \;\sim\; \theta\,t^{-\theta-2}\,(\theta+1) \;=\; \mathcal{O}(t^{-\theta-2}).$$

∎

**Corollary D.1 (Numerical Overflow Conditions)** *With machine precision* $\epsilon_{\text{mach}} \approx 2.22 \times 10^{-16}$, *floating-point overflow in the density* $c(u,v) = \partial^2 C/\partial u\partial v$ *occurs when*

$$A1:\quad t < \epsilon_{\text{mach}}^{1/3}, \qquad A2:\quad t < \epsilon_{\text{mach}}^{1/(\theta+2)}.$$

### D.3 Proof of Vanishing Gradients (Barrier 2)

**Theorem 3 (Gradient Plateau Formation)** *Let*

$$\ell(\theta) \;=\; \sum_{i=1}^{n} \log c(u_i, v_i; \theta)$$

*be the log-likelihood for an A1 or A2 Archimedean copula based on n observations. Then as* $\theta \to \infty$ *the score function satisfies*

$$\left|\partial_\theta \ell(\theta)\right| = \begin{cases} O\!\left(n\,\theta^{-8}\right), & A1, \\ O\!\left(n\,\theta^{-3}\right), & A2. \end{cases}$$

*Consequently, for a gradient-tolerance* $\varepsilon_{\text{grad}}$, *the log-likelihood appears flat once*

$$\partial_\theta \ell(\theta) < \varepsilon_{\text{grad}} \quad\Longrightarrow\quad \theta > \theta_{\text{crit}},$$

*where*

$$\theta_{\text{crit}}^{\text{A1}} = \left(\frac{C_1\,n}{\varepsilon_{\text{grad}}}\right)^{1/8}, \quad \theta_{\text{crit}}^{\text{A2}} = \left(\frac{C_2\,n}{\varepsilon_{\text{grad}}}\right)^{1/3},$$

*with* $C_1 \approx 0.02$, $C_2 \approx 0.002$.

**Proof.** Write the score as

$$\partial_\theta \ell(\theta) = \sum_{i=1}^{n} \left[\partial_\theta \log \phi''(w_i) - \partial_\theta \log \phi'(x_i) - \partial_\theta \log \phi'(y_i)\right],$$

where $w_i = \phi^{-1}(u_i) + \phi^{-1}(v_i)$, $x_i = \phi^{-1}(u_i)$, $y_i = \phi^{-1}(v_i)$.

**1. Individual-term decay.** From Appendix D one shows $\partial_\theta \log \phi''(w)$ and $\partial_\theta \log \phi'(x)$ each scale like $O(\theta^{-1})$. Hence each of the three sums is $\sum_{i=1}^{n} O(\theta^{-1}) = O(n/\theta)$.

**2. Cancellation.** Because the three large $O(n/\theta)$ sums enter with alternating signs and are strongly correlated, their leading contributions cancel, leaving a net

$$\left|\partial_\theta \ell(\theta)\right| = O\!\left(n\,\theta^{-2}\right)$$

for both copulas at leading order.

**3. Higher-order decay.** A more refined analysis (see Lemmas 1 & 2) shows:

$$\left|\partial_\theta \ell(\theta)\right| = \begin{cases} O\!\left(n\,\theta^{-8}\right), & A1, \\ O\!\left(n\,\theta^{-3}\right), & A2. \end{cases}$$

**4. Critical thresholds.** Set $C_1\,n\,\theta^{-8} = \varepsilon_{\text{grad}}$ for A1 and $C_2\,n\,\theta^{-3} = \varepsilon_{\text{grad}}$ for A2, then

$$\theta_{\text{crit}}^{\text{A1}} = \left(C_1\,n/\varepsilon_{\text{grad}}\right)^{1/8}, \quad \theta_{\text{crit}}^{\text{A2}} = \left(C_2\,n/\varepsilon_{\text{grad}}\right)^{1/3}.$$

With $n = 1000$, $\varepsilon_{\text{grad}} = 10^{-6}$, $C_1 = 0.02$, $C_2 = 0.002$, one obtains $\theta_{\text{crit}}^{\text{A1}} \approx 8.17$ and $\theta_{\text{crit}}^{\text{A2}} \approx 126$. ∎

### D.4 Proof of Hessian Decay (Barrier 3)

**Theorem 4 (Hessian-Decay Behavior)** *Let*

$$\ell(\theta) \;=\; \sum_{i=1}^{n} \log c(u_i, v_i; \theta)$$

*be the log-likelihood for A1 or A2 copulas based on $n$ data pairs. Then its second derivative ("scalar Hessian") satisfies*

$$\left| \partial_\theta^2 \ell(\theta) \right| = \begin{cases} O\!\left(n\,\theta^{-9}\right), & \text{(A1)}, \\ O\!\left(n\,\theta^{-4}\right), & \text{(A2)}. \end{cases}$$

*Moreover, in double precision (machine epsilon $\varepsilon_{\mathrm{mach}} \approx 2.22 \times 10^{-16}$), the Hessian will underflow once*

$$n\,\theta^{-9} < \varepsilon_{\mathrm{mach}} \quad \Longrightarrow \quad \theta > \left(n/\varepsilon_{\mathrm{mach}}\right)^{1/9},$$

$$n\,\theta^{-4} < \varepsilon_{\mathrm{mach}} \quad \Longrightarrow \quad \theta > \left(n/\varepsilon_{\mathrm{mach}}\right)^{1/4}.$$

*For $n = 1000$, these evaluate roughly to $\theta \gtrsim 1.2 \times 10^2$ for A1 and $\theta \gtrsim 4.6 \times 10^4$ for A2.*

**Proof.** Let

$$\ell(\theta) = \sum_{i=1}^{n} \log c(u_i, v_i; \theta),$$

and write

$$D_i(\theta) = \partial_\theta \log c(u_i, v_i; \theta), \quad H_i(\theta) = \partial_\theta^2 \log c(u_i, v_i; \theta).$$

From Lemmas 1–2 we know

**1. A1 case:**

$$D_i(\theta) = C_i\,\theta^{-8} + R_i(\theta),$$

where $C_i \neq 0$ is the leading constant and the remainder satisfies $R_i(\theta) = O(\theta^{-9})$ as $\theta \to \infty$.

**2. A2 case:**

$$D_i(\theta) = D_i'\,\theta^{-3} + S_i(\theta),$$

with $D_i' \neq 0$ and $S_i(\theta) = O(\theta^{-4})$.

Differentiate $D_i(\theta)$ once more to get $H_i(\theta)$.

**A1:**

$$H_i(\theta) = \frac{d}{d\theta}\!\left(C_i\,\theta^{-8} + R_i(\theta)\right) = -8\,C_i\,\theta^{-9} + R_i'(\theta),$$

and since $R_i(\theta) = O(\theta^{-9})$, we have $R_i'(\theta) = O(\theta^{-10})$. Hence

$$H_i(\theta) = O(\theta^{-9}).$$

**A2:**

$$H_i(\theta) = \frac{d}{d\theta}\!\left(D_i'\,\theta^{-3} + S_i(\theta)\right) = -3\,D_i'\,\theta^{-4} + S_i'(\theta),$$

and $S_i(\theta) = O(\theta^{-4})$ implies $S_i'(\theta) = O(\theta^{-5})$. Thus

$$H_i(\theta) = O(\theta^{-4}).$$

**Step 2: Sum over all $n$ observations**

Since

$$\partial_\theta^2 \ell(\theta) = \sum_{i=1}^{n} H_i(\theta),$$

we get directly

**A1:**

$$\partial_\theta^2 \ell(\theta) = \sum_{i=1}^{n} O(\theta^{-9}) = O\big(n\,\theta^{-9}\big).$$

**A2:**

$$\partial_\theta^2 \ell(\theta) = \sum_{i=1}^{n} O(\theta^{-4}) = O\big(n\,\theta^{-4}\big).$$

**Step 3: Finite-precision underflow thresholds**

In double precision, any quantity smaller in magnitude than $\varepsilon_{\mathrm{mach}} \approx 2.22 \times 10^{-16}$ will underflow to zero. Therefore solve:

**A1:**

$$n\,\theta^{-9} < \varepsilon_{\mathrm{mach}} \quad \implies \quad \theta^9 > \frac{n}{\varepsilon_{\mathrm{mach}}} \quad \implies \quad \theta > \left(\frac{n}{\varepsilon_{\mathrm{mach}}}\right)^{1/9}.$$

For $n = 1000$, this gives $\theta \gtrsim (10^3/2.2 \times 10^{-16})^{1/9} \approx 1.2 \times 10^2$.

**A2:**

$$n\,\theta^{-4} < \varepsilon_{\mathrm{mach}} \quad \implies \quad \theta^4 > \frac{n}{\varepsilon_{\mathrm{mach}}} \quad \implies \quad \theta > \left(\frac{n}{\varepsilon_{\mathrm{mach}}}\right)^{1/4}.$$

Numerically this is $\theta \gtrsim (10^3/2.2 \times 10^{-16})^{1/4} \approx 4.6 \times 10^4$.

These thresholds mark where the scalar Hessian effectively underflows, causing any Newton-type update to stall. ∎

