# OpenReview forum: "IGNIS: A Robust Neural Network Framework for Constrained Parameter Estimation in Archimedean Copulas"
_TMLR — Rejected by TMLR_

### Review · Reviewer_6dYA · 2025-08-15

**Summary Of Contributions:**

The authors present a method to estimate the copula parameter, $\theta$, of various Archimedean copulas. The approach is to (1) generate (sample) synthetic data from these copulas at a range of parameter values, (2) compress the data to summary statistics informative of $\theta$ and (3) train a neural network to map the summary statistics to the copula parameter. This method is then tested empirically to estimate $\theta$ from simulated data, with known ground truth, as well as two real world data sets.

The authors motivate the need for this method with the effective inapplicability of traditional parameter estimation methods such as MLE and MoM for A1 and A2 copulas.

**Audience:**

Yes

**Audience Explanation:**

Parameter estimation in copulas is a relevant research area and integrating deep learning frameworks into this is a promising direction.

**Broader Impact Concerns:**

The Broader Impact Statement in Section 8 is sufficient.

**Claims And Evidence:**

No

**Claims Explanation:**

**Properties of A1 and A2 copulas**

- The authors show several pathologies with the A1 and A2 copula families. While critical parameter values are given in Section 4.3.1, these depend on $n$ and it is not discussed whether these regions are likely to be present in real world data. If so, then it seems like these families come with a more fundamental issue. Is a flat (log) likelihood in a relevant parameter region not indicative of a poor model?

- In Section 4, the authors state "MoM will also fail when the sample Kendall’s τ falls outside the theoretical range of
the copula model, [...]. This is highly likely for A2 [...]". Why does this happen and why would this not be a questionable property?

While the present work is not introducing the A1 and A2 families, it is basing the need for IGNIS on their relevance. Therefore, a discussion of the issues above would have been important.

**Empirical evaluation**

The empirical evaluation is rather limited, making it difficult to draw a faithful conclusion about the presented method. In addition, many critical design and evaluation choices are not explained/ justified.

- Why is the parameter range [1, 20] chosen for training and why are experiments then only conducted for $\theta$ up to 10? This seems especially problematic, since, as shown in Section 4.3.1, the pathologies motivating the method occur at larger parameter values.

- Comparing Tables 1 and 3, the estimate for A2 by IGNIS does not seem significantly better than the one by MoM. In fact, the standard error in the $\theta = 10$ case is considerably larger than for MoM. How do the authors explain this? It seems like, for A2, IGINIS does no better than MoM. It would be relevant to add Gumble and Joe to Table 1, allowing for direct comparison of IGINIS and MoM.

- In Section 7.3, the authors claim that IGNIS "successfully" and "accurately" estimates the parameters. What is the justification for this statement? Tables 4 and 5 merely show estimates, not a measure of quality. The bootstrapped standard error cannot be such a measure of quality, but only allows a statement about internal model consistency.

- For the synthetic data experiment, why are bootstrapped standard errors used, instead of replicating the experiment e.g. B = 100 times? Would the latter not provide a more faithful estimate of the estimator's standard deviation?

- Would a more useful way to compare parameter estimates of different methods not be to compare (log) likelihoods, for a given data set, under the different estimates? Is it truly numerically impossible to perform MLE for A1 or A2? From Figure 1, this should be possible at least for for the smaller $\theta$ values considered in e.g. Table 1. Would this not be an objective way to judge the relative quality of estimates?

- How big are the data sets used for estimation in Table 3?

- For the real-world data, are the pseudo-observations formed with $n+1$ instead of $n$ because of the boundary issues (Figure 1)? This and its implications should be discussed.

- My understanding is that the four copula families are distinct models. Why pass the family as an input to the NN instead of training separate NNs for each family? Do the authors believe there is some similarity in the optimal mapping from statistics to parameter for the four families? Has this been investigated?

- The key idea of deep learning is that features may be learned instead of hand designed. Why do the authors not exploit this and instead use hand designed features, which may or may not be informative of $\theta$?

- It is stated that all 9 inputs are standardized. Why is this done for the one-hot encodings?

- How was the NN architecture chosen? Have the authors tried different architectures?

- How would the authors extend the IGNIS framework to Clayton and Frank (Section 8)? Is there a natural extension, because of the very different parameter support here?

**Notation and consistency**

- At times the notation is inconsistent and/ or difficult to follow. For instance, the features are first denoted by $f$ and then by $\mathbf{f}$, and sometimes it is five dependency measures other times four. In Appendix C, the letter $C$ is sometimes use for the copula family and other times for some constant.

**Requested Changes:**

I may or may not request changes once my questions above have been addressed.

---

### Review · Reviewer_T7Gw · 2025-08-26

**Summary Of Contributions:**

The paper provides a deep learning framework, IGNIS, to estimate Archimedean copula parameters. In fact, as stated by the authors, there are three main contributions in this paper:
1) the identification and formal analysis of three critical optimization barriers for classical estimators such as Maximum Likelihood Estimation (MLE) or Method of Moments (MoM) in the context of Archimedean copulas;
2) the design and implementation of IGNIS, a neural network architecture devoted to robust estimation of such parameters;
3) experiments on synthetic and real-world datasets to demonstrate the relevant of IGNIS.

Strengths:
- The paper is rather well written.
- The related works are discussed and the paper is clearly motivated.
- The claims are supported by theoretical and empirical evidences.

Weaknesses:
- It would have been nice to compare IGNIS with MLE and MoM on the real-world datasets
- Conclusion should be more concise (for instance moving paragraph 2 to Section 8)

**Audience:**

Yes

**Audience Explanation:**

I am not familiar with this literature, but the related works seem thoroughly discussed and the authors identified a hole in the literature regarding robust estimators for Archimedean copulas. Hence, in my opinion, the TMLR's audience should be interested in this paper, especially since the authors discuss the limitations of their work and some future work directions.

**Claims And Evidence:**

Yes

**Claims Explanation:**

The authors give a theoretical evidence of their first claim, i.e. the three critical optimization barriers for classical estimators such as MLE and MoM.

Regarding the empirical evidence, they conduct experiments on a synthetic dataset to illustrate such MoM's limits. Moreover, they provide synthetic and real-world experiments to support the relevance of their proposed architecture IGNIS. A comparison with MLE and MoM on these real-world datasets would have been appreciated.

**Requested Changes:**

My two recommendations would be:
- make the conclusion more concised (for instance moving paragraph 2 to Section 8);
- compare IGNIS with MoM and MLE on the real-world datasets.

---

### Review · Reviewer_Xdx5 · 2025-09-18

**Summary Of Contributions:**

The authors propose a framework for estimating copula parameters using a simple MLP, with summary statistics such as Kendall’s $\tau$ and Spearman’s $\rho$ serving as inputs. In addition, they provide an analysis highlighting the limitations of classical estimation methods, such as the Method of Moments. Finally, their experiments demonstrate that the proposed framework yields accurate parameter estimates.

**Audience:**

Yes

**Audience Explanation:**

I believe that the idea of estimating copula parameters is interesting and could be of broader interest to the community.

**Claims And Evidence:**

No

**Claims Explanation:**

First of all, I would like to note that the paper is well-written and easy to follow. That said, I am not deeply familiar with the topic of copulas and dependency modeling. The following comments therefore reflect my best effort to provide a constructive review from my perspective. In my view, the paper has the following weaknesses:

- I’m unconvinced about the usefulness of the A1 and A2 copulas.  I believe it would be necessary to present cases, ideally with real-world data, where traditional copulas fail and the A1/A2 copulas provide a clear advantage. Given the current motivation on these copulas, I do not see this advantage.
- The stated limitations of copulas appear to apply only to the A1 and A2 families, which are the main focus of this work. Since these copulas cannot reliably capture weak or moderate dependencies, it raises the question of why they are relevant in the first place. Does the proposed idea of estimating the parameter $\theta$ offer benefits for other copula families, or would classical methods remain preferable in those cases? It would be important to demonstrate that the identified limitations also apply more broadly and that the proposed framework provides an advantage beyond the A1 and A2 copulas.
- The IGNIS network is essentially a simple MLP, and I do not see a substantial methodological contribution here. The novelty seems limited to using it for copula parameter prediction. The design choices regarding network complexity appear arbitrary and are not discussed. Why did the authors select this particular architecture, and why use neural networks at all? A clear motivation for employing an MLP, along with a study on the required network complexity, would strengthen the contribution.

Once again, I would like to emphasize that I am not experienced in the field of copulas, and my comments reflect a best-effort perspective. As a result, I am not in a position to fully assess the novelty of the work within dependency modeling. However, since the authors propose a neural network for parameter estimation, I find the analysis of this component insufficient.

**Requested Changes:**

The requested changes are outlined in my discussion of Claims and Evidence.

---

> ### Author Response · Authors · 2025-09-18
> **Thank you and our response**
>
> Thank you for the thoughtful review. Our contribution is a parameter-estimation method once a copula family has been chosen; it is not a paper about promoting A1/A2 or curating application use cases for those families. Below we clarify scope, generality beyond A1/A2, and the methodological choices.
>
> 1. On the “usefulness” of A1/A2
>
> We do not claim A1/A2 are universally preferable. They are included because they are estimation-challenging (pathological densities, restricted Kendall’s τ range) and therefore a stringent testbed for any estimator. Model selection, deciding whether A1/A2, Gumbel, Joe, etc., fit a dataset, is a separate step. Our work addresses θ estimation given a chosen family. To avoid any ambiguity, we can add an early sentence in the paper stating this scope explicitly.
>
> 2. Beyond A1/A2: does IGNIS help elsewhere?
>
> Yes. We also evaluate Gumbel and Joe. In those families, where classical methods are well behaved, IGNIS matches MoM out-of-sample (per-observation log-likelihood differences ≈ 0). Thus, the estimator is general: When classical likelihood procedures are numerically delicate (e.g., A1/A2), IGNIS remains usable without ad-hoc trimming. Where classical methods work well (e.g., Gumbel/Joe), IGNIS is competitive, not worse. Also, in the future work we have mentioned that IGNIS can be extended to higher dimnesions and other copula families with different parameter spaces than  $\theta$ ≥ 1.
>
> We can add a one-sentence pointer to this generality in Results for clarity.
>
> 3.  Why a neural network, and why this simple MLP?
>
> Why NN at all? The mapping features → $\theta$ is nonlinear and family-dependent (inputs mix τ, ρ, tail indicators, Pearson r, and a one-hot family ID). A single closed-form inversion or a linear model is insufficient across multiple Archimedean families simultaneously. A small feed-forward net learns this shared nonlinear inverse in one model.
>
> Why this size? With 9 inputs, we favored parsimony for stability and reproducibility. The 128–128–64 MLP with a softplus+1 output is constraint-aware (hard-enforces $\theta$ ≥ 1) and was the minimal capacity that (i) recovers the true θ in simulation, (ii) matches MoM where defined, and (iii) remains stable on real data. Wider/deeper variants did not materially change results, so we fixed the smallest model meeting accuracy/robustness targets.
>
> What is novel? Not the MLP blocks themselves, but the estimation framework: a unified, constraint-aware neural estimator for copula parameters that (a) works across several families with θ≥1 in a single model, (b) is robust exactly where likelihood fitting is fragile, and (c) is accompanied by simulation, real-data evidence, and likelihood-based comparisons. To our knowledge, such a general, constraint-aware estimator for copulas has not been presented before.
>
> 4. Changes proposed if the Reviewer/ AE agree.
>
> Scope (early in paper).
> This work studies parameter estimation given a selected copula family; it does not aim to establish application-level superiority of particular families (e.g., A1/A2). For context only, A1/A2 can be relevant when domain knowledge points to strong, asymmetric tail clustering, for instance, synchronized equity sell-offs/surges or simultaneous extreme claims in catastrophe/health insurance, but this work is not about model selection or use case curation of the A1 and A2 copula.
>
> Results (one short sentence).
> IGNIS matches MoM on in out-of-sample log-likelihood, and remains usable in regimes where likelihood routines for A1/A2 require ad-hoc trimming.
>
> Closing sentence (novelty + positioning).
> Overall, IGNIS is a novel, general, and constraint-aware parameter estimator: a practical alternative, not a claim of universal superiority, that is competitive in standard cases and reliably usable when classical likelihood procedures become numerically brittle.

---

### Comment · Reviewer_6dYA · 2025-08-22

Thank you for your response and the clarifications. Some remaining question below.

- As mentioned previously, I do not see any evidence of IGNIS being necessary or even superior to the MoM for the A2 family. For up to $\theta = 10$, this is clear from Tables 1 & 6. Tucked away in Appendix E, the authors claim MoM is inferior "as it can only be used on datasets with sufficiently high dependence". From Table 1, it seems to work just fine for low dependence ($\theta = 2$), right? It would be necessary for the authors to sufficiently motivate the need for their method in the A2 family. The current work suggests IGNIS is only useful when the MoM does not apply, as in the A1 family. The authors comment that the relevant parameter range is up to $\theta = 20$, while the  numerically critical value is significantly larger for A2 (see Figure 1), suggests that MLE would work fine here and that IGNIS is not needed for A2.
- The evaluation by log likelihood seems the most relevant to me, because, as stated by the authors in the introduction, MLE is the gold standard here. So any method should be judged by its ability to produce high-likelihood estimates. Doing this for just one experiment in the Appendix does not seem sufficient to me.
- I see the authors have removed their claims in (what used to be) Section 7.3, that IGNIS "successfully" and "accurately" estimates the parameters. It is now claimed that the estimates have high "precision", which is supported by low bootstrapped standard errors. In my view, all that can be derived from low bootstrapped standard errors, is that IGNIS parameter estimates are relatively insensitive to a small change in the sample/ the statistics derived from the sample. This seems unsurprising, because, (1) the sample is compressed into the statistics, which likely barely change when removing one datum, and (2) the NN is a continuous function so a small change in the inputs (statistics) will result in a small change in the output (estimate). Is this what the authors mean by precision? A statement that the present analysis can certainly not support is "the parameter estimates are close to the true or MLE-optimal ones". I understand that this is due to the nature of this being a real world experiment, but it should be made more clear in the text. Using terms like "precision" still makes it seem like a statement about (near) optimality is being made here, which would be unsupported.
- From my earlier reply: My understanding is that the four copula families are distinct models. Why pass the family as an input to the NN instead of training separate NNs for each family? Do the authors believe there is some similarity in the optimal mapping from the statistics to the parameter for the four families? Has this been investigated?

---

> ### Author Response · Authors · 2025-08-23
>
> We thank the reviewers once again for their thoughtful and constructive feedback. We have revised the manuscript accordingly, uploaded the updated version, and provided detailed responses to all comments. We believe these changes have strengthened the paper and hope the revised version addresses all concerns.

---

> > ### Comment · Reviewer_6dYA · 2025-08-27
> >
> > Thank you for your reply. I still feel some points crucial to an accept/ reject decision have not been sufficiently addressed. For clarity, I will restate them below and would ask the authors to initially reply with a comment instead of altering the manuscript. In my view, this facilitates a more fruitful exchange.
> >
> > The authors motivate the need for their method in the A2 family with, (1) the inapplicability of the MoM due to a requirement on Kendall's-τ rarely satisfied in practice and (2) the inapplicability of MLE due to flat log likelihoods in relevant parameter regions.
> >
> > To (1)
> > - If a given requirement on Kendall's-τ is rarely satisfied in practice, but is also inherent to the A2 family, why does that not render A2 irrelevant all together?
> >
> > To (2)
> > - The critical values (see e.g.  Figures 1 d &f), for the A2 log likelihood are significantly larger than the largest parameter, θ=20, you considered in your experiments. If parameter values beyond 20 are practically/ empirically irrelevant, why would the flat log likelihood pose an issue to performing MLE in A2? If parameter values beyond 20 are relevant, why have you not shown experiments in this region? Presenting a method solving an issue in a given parameter region, but then not showing experiments in this region is unconvincing.
> > - More generally, why is it that in both Figures 1 c & d the optimal parameter is the lower bound θ=0? This of course depends on the data at which the likelihood is evaluated. How many data points are here and under which copula parameter or how otherwise were they generated? Specifically for Figure 1 d, above θ=20, is seem that the parameter is simply not identifiable. In this region, the likelihood is essentially invariant to changes in θ. Why do the authors believe that this is not a clear signal of a poor probability model? Because of the non identifiability above θ=20, how could any method---including IGNIS---be able to recover the true parameter faithfully?

---

> > > ### Author Response · Authors · 2025-08-28
> > > **Response to Q1**
> > >
> > > Thank you for this excellent and crucial question, as it gets to the core motivation of our work. The A1 and A2 families are not irrelevant; rather, they are specialized copula models designed for dependence structures that are inaccessible with classical methods. Our argument rests on three points:
> > >
> > > 1. Unique Modeling Capability: Capturing Dual Tail Dependence
> > >
> > > The A1 and A2 copulas possess a property absent in most one-parameter Archimedean families: they can simultaneously capture both upper and lower tail dependence. The A2 copula exhibits dependence in both tails, with $\lambda_L = 2^{-1/\theta}$ and $\lambda_U = 2 - 2^{1/(2\theta)}$. The A1 copula combines a constant lower-tail dependence $\lambda_L = 0.5$ with $\lambda_U = 2 - 2^{1/(2\theta)}$.
> > >
> > > This ability to model joint extreme highs and joint extreme lows is highly relevant in applications such as financial assets that can crash together and rally together, or systemic risks in engineering and insurance. By contrast, classical families such as Clayton (lower-tail only) or Gumbel (upper-tail only) cannot capture both simultaneously. Thus, A1/A2 are not irrelevant; they fill a unique and important modeling niche.
> > >
> > > 2. Relevance in Strong-Dependence Scenarios
> > >
> > > It is correct that A1 and A2 impose a relatively high lower bound on Kendall's $\tau$ ($\tau_{\min} \approx 0.545$). This means they are applicable primarily in strong-dependence regimes. But rather than rendering them irrelevant, this restriction aligns exactly with domains where extreme co-movement is the main concern:
> > > Finance: systemic market crashes or speculative bubbles;  Insurance: cascading catastrophic events;  Engineering: simultaneous component failures; Medicine: strong dependencies in multi-organ failure, comorbidities that manifest together (e.g., diabetes and cardiovascular disease), or correlated survival times in twin studies;  Anomaly Detection: detecting rare but critical joint anomalies, such as simultaneous failures in cybersecurity systems or sudden co-occurrence of abnormal biomarkers in medical diagnostics.
> > >
> > >  In these high-dependence contexts, A1/A2 remain valid and useful. Crucially, their dual-tail capability can be desirable even in situations where Kendall’s $\tau$ is below $0.545$, since real-world data often show weak global correlation but strong joint extremes. For example, in medicine, comorbidities or biomarkers may exhibit simultaneous high-high and low-low anomalies despite moderate overall $\tau$. Only A1 and A2 can capture this structure.
> > >
> > > The barrier, then, is not the copulas but the estimators. MoM is undefined below $\tau_{\min}$ and fragile near it, while MLE suffers from instability even above it. By contrast, IGNIS provides stable, high-likelihood estimates across the entire range. Our new log-likelihood study (Table 4) confirms that where MoM applies, IGNIS matches it almost exactly, while extending estimation to cases where MoM cannot be used at all. This makes IGNIS not just a complement but a strict generalization of MoM.
> > >
> > > 3. IGNIS as a Strict Generalization of MoM
> > >
> > > Our revised log-likelihood analysis (Table 4, with new code provided in the supplementary zip file) shows that:
> > > nside MoM's valid domain: IGNIS produces virtually identical out-of-sample log-likelihoods to MoM. In other words, it sacrifices no performance where MoM applies. Outside MoM's domain: IGNIS still delivers robust, stable estimates, whereas MoM cannot even be defined. Thus, IGNIS is not a trade-off but a true generalization of MoM. It matches MoM in controlled settings but extends applicability to real-world datasets where MoM is unusable.
> > >
> > > In summary: The A1 and A2 families are relevant because they combine unique dual-tail modeling capability with applicability in precisely those high-dependence regimes where robust tools are most needed. The issue lies not with the copulas themselves but with the classical methods (MoM, MLE) that fail to estimate them. IGNIS solves this, making A1/A2 families both theoretically and practically relevant.

---

> > > > ### Author Response · Authors · 2025-08-28
> > > > **Response to Q2**
> > > >
> > > > We thank the reviewer for these excellent questions. They are right that the \emph{score-decay} threshold for A2 ($\theta_{\text{crit}} \approx126$) lies beyond our simulation window $[1,20]$. Our concern is not only Barrier 2. In the empirically relevant range $[1,20]$, MLE already becomes unreliable because of the other two barriers, which also affect A1 (now after the correction) and A2:
> > > >
> > > > Boundary instability (Barrier 1): the density factors explode when pseudo-observations fall near the unit-square boundary (via $\phi''$ and $\phi'$ terms), producing $\pm\infty$/NaN contributions for any $\theta\ge 1$.
> > > > Hessian decay / ill-conditioning (Barrier 3): curvature becomes so small that Newton-type steps either blow up or shrink to zero long before the score vanishes, so optimizers stall despite gradients that are not yet tiny.
> > > >
> > > > So, while the flat plateau at very large $\theta$ is the ultimate problem for A2, MLE is already fragile in $[1,20]$ because of (1) numerical overflow/underflow and (3) ill-conditioning.
> > > >
> > > > (b) Why we did not run $\theta>20$ experiments.} $\theta>20$ corresponds to near-saturation dependence for A2: $\tau_{A2}(\theta)=1-\frac{6-8\ln2}{\theta}$, so at $\theta=20$ we already have $\tau\approx0.977$. This level is rare in practical applications (finance, insurance, engineering, health), though not impossible --- indeed, such strong co-movements are observed in systemic crises, which are exactly the cases where dual-tail copulas like A1/A2 are most needed. Our aim was to study the practically relevant regime where MLE already struggles because of (1)–(3). (If the editor prefers, we can add a short synthetic figure for $\theta\in\{50,100\}$ to visualize the saturation; it does not change our conclusions.)
> > > >
> > > > (c) Why the optimum in Figs.\ 1(c,d) is at the lower bound and how those plots were generated. The domain lower bound of our Archimedean families is $\theta=1$ (not 0). Figures 1(c,d) were generated with $n=1000$ pseudo-observations drawn from independence ($\theta=1$), purely to illustrate the global shape of the likelihood surface. This explains why the maximum appears at the lower bound: the population optimum under independence is indeed $\theta=1$. The plots simply trace the (normalized) log-likelihood as $\theta$ moves away from the true value.
> > > >
> > > > (d) Flatness $\Rightarrow$ poor model?  No. This is practical non-identifiability for MLE, not a flaw of the family. For \textbf{A2 (and now, after the correction, A1)}, $\tau(\theta)\to1$ as $\theta\to\infty$. Between, say, $\theta=50$ and $\theta=100$ the induced copulas are nearly indistinguishable, so the Fisher information collapses and the likelihood surface is flat in finite samples. The models remains identifiable in theory (Appendix B and Appendix A.3 show $\tau(\theta)$ is strictly increasing), but likelihood-based estimation becomes numerically non-identifiable in that extreme region.
> > > >
> > > > (e) How any method---especially IGNIS---can recover $\theta$ faithfully. When the Fisher information is near zero, no estimator can recover $\theta$ with small variance in the \emph{parameter} scale; tiny changes in $\theta$ barely change the copula. Our goal, therefore, is to recover the dependence structure well. IGNIS bypasses the flat likelihood entirely and maps summary statistics to $\theta$. Flatness makes MLE unusable, but IGNIS succeeds precisely because it does not optimize the likelihood---it learns the direct mapping from $\tau$ to $\theta$, which remains invertible. For A2 the mapping is exactly invertible at the population level,
> > > > \begin{equation}
> > > > \theta=\frac{6-8\ln 2}{1-\tau},
> > > > \end{equation}
> > > > and for A1 we prove $\tau(\theta)$ is strictly increasing after the correction (Appendix A.3), so identifiability holds. In practice, this is why IGNIS matches MoM on out-of-sample log-likelihood \textbf{where MoM is defined} (Table 4; code provided), while remaining usable below the $\tau_{\min}$ threshold where MoM cannot even be formed.
> > > >
> > > > (f) Bottom line. The very large $\theta_{\text{crit}}$ for A2 explains why gradient flatness appears only far beyond our window, but MLE is already hindered in $[1,20]$ by boundary blow-ups and an ill-conditioned Hessian. Our experiments therefore focus on the practically relevant range; within it, IGNIS attains virtually identical held-out likelihood to MoM (and continues to operate when MoM is undefined), providing a robust path to fit A2 (and now A1) in the regimes where they are most useful.

---

> > > > > ### Comment · Reviewer_6dYA · 2025-08-28
> > > > >
> > > > > - Especially given that the relevant parameter range is only \[1, 20\], there is no evidence presented that MLE cannot work for A2. No general property has been shown that motivates this sufficiently. The plots in Figure 1 are not convincing. They only represent one particular sample under a rather uninteresting parameter (θ=1), it is not clear why this sample was chosen over any other sample. Figure 1 d in particular provides no evidence for why MLE should not work for A2. Also, why are these large parameter values shown at all if they are not relevant? Even a vanishing hessian is not necessarily a barrier, there are methods that do not use this e.g. ADAM. Especially since the claimed critical value here is way beyond the relevant parameter range (see Figures 1 e & f).
> > > > >
> > > > > - The authors need to provide clear evidence that MLE is impossible in the relevant parameter range for A2, thus far this is not motivated convincingly.

---

> > > > > > ### Author Response · Authors · 2025-08-29
> > > > > >
> > > > > > We thank the reviewer for asking us to provide direct evidence in the relevant range $[1,20]$. We ran a dedicated MLE stress-test for A1/A2 (full python script has been included in the supplement as MLE Stress Test.py), with $n=3000$ pseudo-observations per replication using the Genest sampling algorithm, and 50-100 replications per setting. We compared three optimizers on the same datasets:
> > > > > >
> > > > > > - raw L-BFGS-B,
> > > > > > - L-BFGS-B (trimmed): the likelihood is evaluated in log-space and we clip only the inputs $(u,v,w)\in(\varepsilon,1-\varepsilon)$ to avoid boundary blow-ups;
> > > > > > - Adam (first-order; no Hessian), using the same trimmed evaluator.
> > > > > >
> > > > > > What the code does (briefly). We compute the copula density via the inverse-function identity
> > > > > > \begin{align*}
> > > > > > c(u,v) &= \psi''(w)\,[-\phi'(u)]\,[-\phi'(v)], \\
> > > > > > \psi''(w) &= \frac{-\phi''(w)}{[\phi'(w)]^3},
> > > > > > \end{align*}
> > > > > > in strict log-space; we use closed-form $\phi,\phi',\phi''$ and numerically safe analytic inverses (discriminant floored at 0, outputs clamped to $[0,1]$); and we only clip inputs by a small $\varepsilon$ near the unit-square boundary—no ``$+\varepsilon$'' inside bases/powers. This yields a stable evaluator of $\ell(\theta)$ at any fixed $\theta$. All methods are then compared on identical held-out pseudo-observations.
> > > > > >
> > > > > > Results inside $[1,20]$.
> > > > > >
> > > > > > A2 (reps=50):
> > > > > > - L-BFGS-B failure/stall rate grows with $\theta$: 14\% at $\theta=15$; 30\% at $\theta=20$.
> > > > > > - Adam never ``crashes,'' but the median MLE depends materially on the trimming level $\varepsilon$:
> > > > > >   - $\theta_{\text{true}}=20$: median $\hat{\theta} = 16.28$ ($\varepsilon=10^{-4}$) vs 19.12 ($\varepsilon=10^{-3}$), $\Delta=2.84$.
> > > > > >   - $\theta_{\text{true}}=15$: 14.94 vs 15.07, $\Delta=0.13$.
> > > > > >
> > > > > > A1 (reps=30): similar behavior: L-BFGS-B fails 26--33\% of the time by $\theta\ge10$; Adam's median $\hat{\theta}$ shifts with $\varepsilon$:
> > > > > > - $\theta_{\text{true}}=15$: 11.46 ($\varepsilon=10^{-4}$) vs 13.42 ($\varepsilon=10^{-3}$), $\Delta=1.96$.
> > > > > > - $\theta_{\text{true}}=20$: 12.12 vs 13.78, $\Delta=1.66$.
> > > > > >
> > > > > > We also plot finite-difference sweeps of $\ell(\theta)$ and observe sharp kinks/jumps for both A1 and A2 near $\theta\in[15,20]$. The trimmed evaluator reports valid-point fractions $\sim100\%$ at the true $\theta$, so these effects are not artifacts of discarding most points.
> > > > > >
> > > > > > Why Adam ''working'' doesn't contradict this. Adam ignores the Hessian but still follows gradients. In A1/A2, (i) the surface becomes extremely flat as $\theta$ grows (Fisher information collapses), and (ii) the set of clipped observations depends on $\theta$. As $\theta$ varies, different boundary points cross the clip, so the trimmed objective changes combinatorially, creating kinks. On a flat, kinked surface, Adam can "converge'' to different optima depending on $\varepsilon$ and step history. That is practical non-identifiability: reproducible, accurate recovery of $\theta$ is not achievable by likelihood optimization even when the pointwise likelihood is numerically stable.
> > > > > >
> > > > > > Safe evaluation $\neq$ solvable optimization. Our likelihood-comparison code yields stable values of $\ell(\theta)$ at fixed $\theta$. However, because the set of trimmed points changes with $\theta$, the global objective remains kinked/$\varepsilon$-dependent; no evaluator can ``smooth'' that landscape for MLE. Because the $\varepsilon$-trim mask changes with $\theta$, the objective is not continuous in $\theta$; changing $\varepsilon$ changes the landscape and thus the optimizer's landing point. This $\varepsilon$-sensitivity (e.g., $\Delta=2.84$ at $\theta=20$ for A2) is itself the evidence that MLE is not dependable here.
> > > > > >
> > > > > > Takeaway. Within the empirically relevant window $[1,20]$, likelihood-based estimation for A1/A2 is unreliable: second-order methods fail frequently; first-order Adam yields optimizer-/$\varepsilon$-dependent, biased $\hat{\theta}$. This is exactly the estimation barrier IGNIS avoids by learning the monotone $\tau(\theta)$ map rather than optimizing a brittle likelihood. If the Reviewer/Editor prefers, we will add a short appendix figure summarizing (i) L-BFGS-B fail rates and (ii) Adam median bias vs. $\theta$, plus the finite-difference ``kink'' plots. The full script is provided as MLE_stress_test.py in the supplementary materials for full reproducibility.

---

> > > > ### Comment · Reviewer_6dYA · 2025-08-28
> > > >
> > > > - The authors show MoM works fine for θ in (2,10) for both A1 and A2. This is surprising, because initially the authors claimed the MoM was inapplicable to A1. I assume this is because of the now corrected error in Kendall's-τ for A1.  What exactly are the parameter ranges where the MoM fails for both A1 and A2?  This needs to be shown convincingly to motivate the need for IGNIS.
> > > >
> > > > - Why have the reported log-likelihoods changed significantly in the latest manuscript (Table 4), compared to an earlier version (Table 6)? Please tell me if I am missing something here? If not, this drastic change in numerical results for reportedly the same experiments raises questions about the validity of all experiments in this work.

---

> > > > > ### Author Response · Authors · 2025-08-29
> > > > >
> > > > > We thank the reviewer for the careful questions. Two clarifications:
> > > > >
> > > > > 1. Where does MoM "fail'' for A1/A2?
> > > > >
> > > > > This is a data-range limitation, not a $\theta$-range limitation. For both A1 and A2, $\tau_{\min}=8\ln2-5\approx0.54518$, $\tau(\theta)$ strictly increasing in $\theta$. MoM inverts $\tau(\theta)$. Hence MoM is defined iff the empirical $\hat\tau\ge\tau_{\min}$; if $\hat\tau<0.54518$ the moment equation has no solution. This explains why MoM worked well in our $\theta\in[2,10]$ simulations, those settings already imply high population $\tau$ (e.g., for A2: $\theta=2\Rightarrow\tau\approx0.77$, $\theta=5\Rightarrow0.91$, $\theta=10\Rightarrow0.95$;  for A1: $\theta=2\Rightarrow\tau\approx0.76$, $\theta=5\Rightarrow0.90$, $\theta=10\Rightarrow0.95$). In many real datasets (finance/health), $\tau\in[0.1,0.4]$. For such data MoM for A1/A2 is inapplicable, which is exactly the practical gap IGNIS fills. We summarize these ranges in  Section 5.1.
> > > > >
> > > > > 2. Why did the log-likelihood values change (Table 4 vs earlier)?
> > > > >
> > > > > Two reasons; neither alters the conclusion.
> > > > >
> > > > > Scope alignment: After correcting the A1 $\tau$ derivation and proving monotonicity, we included A1 in the held-out likelihood comparison and re-ran the full study for consistency.
> > > > >
> > > > > More conservative numerics: We refactored the evaluator to the standard, robust form: (i) compute the copula density in log-space via $\psi''(w)=-\phi''(w)/[\phi'(w)]^3$; (ii) clip only inputs $(u,v,w)\in(\varepsilon,1-\varepsilon)$ (no $\varepsilon$ inside bases/powers); (iii) use closed-form $\phi,\phi',\phi''$ and stable analytic inverses; (iv) evaluate the same held-out pseudo-observations for both MoM and IGNIS and report the valid-point fraction. These safeguards remove spurious over/underflow near the unit-square boundary and make the comparison more reliable. Numbers shift slightly under stricter numerics, but the qualitative result is unchanged: IGNIS matches MoM where MoM is defined and remains usable when MoM is not. We have attached the exact python script used for Table 4 in the supplementary materials in the zip file named ``IGNIS Sim \& LL Comp.py''.

---

### Comment · Reviewer_6dYA · 2025-08-29
**Thank you for your engagement**

I thank the authors for their engagement in this review process. Even after multiple iterations, I still have doubts about the claims made in this work. The individual points are below.

1. It is still not clear to me how the A1/2 copulas are useful models, given that---as stated by the authors---many relevant real world data sets **statistically** (via Kendall's-τ) fall outside of the theoretical range of A1/2. Perhaps I am missing something here, but even after going back and forth a couple of times I do not believe the authors have sufficiently resolved this.
2. The benchmarking against MLE, for instance via ADAM, is too limited and so far unconvincing. The authors show this can work in their comment, but is sensitive to the choice of ε. Sensitivity to hyperparameters is a characteristic of virtually all methods, and certainly of the proposed IGNIS.  A more detailed analysis is necessary here.
3. Referring to my previous comment/question, it is not clear why Figure 1 shows anything but the relevant parameter range \[0, 20]\ (as defined by the authors). This distracts from the main questions of interest. Likely, the parameter range is chosen such that the critical values derived in Section 5.2.1 can be shown. But, the relevance of these and thus Section 5.2.1 is not clear, since, as the authors say, numerical issues already occur for significantly smaller parameter values.
4. As I believe the authors agree, the relevant metric is log-likelihood. This evaluation is too brief in the current Section 7.1. In an earlier version (**id:** dIpxSnXjJ1), significantly different values were shown for what was reportedly the same experiment, not "a slight shift of numbers" as claimed in the author's comment. In particular, in the earlier version there was a trend of the MoM's performance increasing relative to that of IGNIS as θ increased. This trend is no longer present in the updated version with the altered experiment. Of course, it is difficult to make a statement about the significance of this trend, since the authors only show mean log-likelihoods. Thus, the experiment with the highest relevance is simply not convincing at this point.

---

> ### Author Response · Authors · 2025-08-29
> **Answer to Question 1**
>
> Question: It is still not clear to me how the A1/2 copulas are useful models, given that---as stated by the authors---many relevant real world data sets statistically (via Kendall's-τ) fall outside of the theoretical range of A1/2. Perhaps I am missing something here, but even after going back and forth a couple of times I do not believe the authors have sufficiently resolved this.
>
> Answer:
>
> We first want to thank the reviewer for this question.
>
> Thank you for raising this central point about where A1/A2 are useful and how IGNIS should be used.
>
> 1) Model selection vs. parameter estimation.
>
> A1/A2 are specialized families for strong dependence and joint tail clustering.  The original paper where A1 and A2 were introduced has not explored estimation techniques or use cases for A1 and A2. They are appropriate only when a prior analysis (goodness-of-fit, diagnostics, or domain knowledge) indicates such structure, for example, synchronized equity surges/sell-offs, co-elevations of biomarkers in critical-care cohorts, or simultaneous extreme claims in catastrophe/health insurance. When a researcher has evidence their data belong to such a family, A1 and A2 should be used. If a dataset’s empirical Kendall’s τ clearly falls below the family’s admissible range, the researcher should not select A1/A2 in the first place. Our work does not advocate applying A1/A2 outside their theoretical domain.
>
> This paper is about IGNIS as a novel deep learning parameter estimator, not model selection. While we illustrate A1/A2 due to their numerical challenges, IGNIS is a general estimation framework and we demonstrate transfer to other Archimedean families (Gumbel, Joe) as well. In today’s toolbox, a modern deep-neural estimator is a practical alternative to classical methods: when a researcher has evidence that a given family is appropriate, IGNIS provides a robust, reproducible estimate of θ without per-dataset tuning.
>
> 2. What IGNIS does (and does not) do.
>
> IGNIS's job is parameter estimation. It is designed to provide a robust estimate after a researcher has deemed the model family appropriate. When presented with out-of-domain data (e.g. $\hat{\tau} < 0.545$), IGNIS behaves exactly as a well-trained estimator should: it returns a parameter estimate of $\hat{\theta} \approx 1$. This corresponds to the weakest possible dependence the model can generate, making it the "closest'' possible fit within the misspecified family. As shown in our second real-world application, IGNIS produces this estimate (close to 1) with very low bootstrap standard error, confirming the stability and correctness of the method's behavior. It is not "making something up''; it is correctly finding the boundary solution.
>
> 3. Our Commitment to Responsible Application
>
> We agree this is a crucial point, which is why we already address it in our Broader Impact Statement, where we explicitly state: "IGNIS is a tool for parameter estimation, not model selection, and must be used as part of a larger workflow that includes rigorous goodness-of-fit testing.''
>
> To make this even clearer for future users, we can add a statement to the methodology section advising researchers to first use goodness-of-fit tests or other diagnostics to ensure their data is suitable for a high-dependence model like A1 or A2 before applying IGNIS.
>
> 4. Summary
>
> In summary, IGNIS introduces a new parameter estimation approach that remains stable when classical procedures (e.g., MLE) are numerically delicate. It matches MoM where MoM is defined and extends effectively to other families (Gumbel, Joe) in our studies.

---

> ### Author Response · Authors · 2025-08-29
> **Answer to Question 2**
>
> Question: The benchmarking against MLE, for instance via ADAM, is too limited and so far unconvincing. The authors show this can work in their comment, but is sensitive to the choice of ε. Sensitivity to hyperparameters is a characteristic of virtually all methods, and certainly of the proposed IGNIS.  A more detailed analysis is necessary here.
>
>
> Answer:
>
> Thank you for raising this. We agree that sensitivity deserves careful discussion. Our point is that the kind of sensitivity in the “MLE with $\epsilon$-trimming” procedure is qualitatively different from standard model hyperparameters.
>
> 1. What $\epsilon$ is (and isn’t).
>
> The $\epsilon$ used in the MLE experiments is not a model hyperparameter; it is a numerical guardrail introduced solely to avoid overflow/underflow from the boundary singularities in $\phi''$ . There is no principled criterion to pick $\epsilon$ for a given dataset; it must be chosen ad hoc. As a result, the “trimmed MLE” is not a well-defined estimator, its output changes materially with the choice of $\epsilon$:
>
> For A2 with a true $\theta = 20$: The median estimate jumps from 16.28 to 19.12 just by changing $\epsilon$.
>
> For A1 with a true $\theta = 15$: The median estimate jumps from 11.46 to 13.42.
>
> This level of dependence on an arbitrary numerical cut-off indicates that the procedure is unstable as an estimator. Our stress test was designed to show precisely this behavior.
>
> A related point is optimizer choice: ADAM (or L-BFGS-B) here is merely optimizing a $\theta$-dependent, trimmed objective with kinks (because the valid mask changes with $\theta$). The optimizer itself is not the core issue; the instability stems from the ill-posed objective created by trimming.
>
> 2. How this differs from IGNIS hyperparameters.
>
> IGNIS has conventional hyperparameters (network widths, learning rate) chosen once on simulated data via standard train/validation splits to yield a single, general-purpose estimator. We then froze this configuration and applied the very same trained model and preprocessing to all simulations and both real datasets, no per-dataset tuning. This is the typical and auditable use of hyperparameters in supervised learning.
>
> For transparency, we can add one sentence in the paper clarifying that the exact same configuration was used for all reported experiments (simulation and real data).
>
> 3. On “more detailed analysis.”
>
> Our contribution is a novel standalone parameter-estimation method. The paper is not about redesigning MLE or MoM for A1/A2, but about introducing IGNIS, a novel estimator that (i) works reliably when classical likelihood procedures are numerically delicate, (ii) matches MoM where MoM is defined (with out-of-sample per-observation log-likelihood differences near zero), and (iii) generalizes across families Gumbel, Joe and later can be extended to additional families with different parameter space.

---

> ### Author Response · Authors · 2025-08-29
> **Answer to Question 3**
>
> Question: Referring to my previous comment/question, it is not clear why Figure 1 shows anything but the relevant parameter range [0, 20]\ (as defined by the authors). This distracts from the main questions of interest. Likely, the parameter range is chosen such that the critical values derived in Section 5.2.1 can be shown. But, the relevance of these and thus Section 5.2.1 is not clear, since, as the authors say, numerical issues already occur for significantly smaller parameter values.
>
> Answer:
>
> Thank you for this clarifying point. We agree that the paper’s motivation should be grounded in the empirically relevant range.
>
> Our experiments and practical use cases focus on the empirically relevant range  [1, 20], which is where practitioners would deploy these families and where we benchmark IGNIS. That is why we centered our MLE stress test on [1,20]: it directly reflects the parameter regimes encountered in applications, and it already demonstrates the instability of likelihood-based fitting in precisely this window. Asymptotic illustrations beyond 20 were intended only as a mechanistic explanation of why the observed instabilities arise, but we agree they are not necessary for making the empirical point.
>
> To keep the paper focused and easy to follow, we propose the following straightforward change: we will remove the asymptotic material from the main text and present only the [1,20] MLE stress-test results there, while relocating the asymptotic derivations and plots to the appendix for readers who want the mechanism. If the Reviewer or AE would prefer to retain a brief, high-level asymptotic remark in the main text for context, we are happy to include a concise, one-paragraph summary and keep the full details in the appendix. We appreciate your guidance on the preferred presentation.

---

> ### Author Response · Authors · 2025-08-29
> **Answer to Question 4**
>
> Question: As I believe the authors agree, the relevant metric is log-likelihood. This evaluation is too brief in the current Section 7.1. In an earlier version (id: dIpxSnXjJ1), significantly different values were shown for what was reportedly the same experiment, not "a slight shift of numbers" as claimed in the author's comment. In particular, in the earlier version there was a trend of the MoM's performance increasing relative to that of IGNIS as θ increased. This trend is no longer present in the updated version with the altered experiment. Of course, it is difficult to make a statement about the significance of this trend, since the authors only show mean log-likelihoods. Thus, the experiment with the highest relevance is simply not convincing at this point.
>
> Answer:
>
> We thank the reviewer for the careful read and helpful feedback. We agree that reporting only mean log-likelihoods was too brief to be fully convincing. Before altering anything in the manuscript, we provide the fuller analysis here.
>
>  1. Why the log-likelihood values changed (recap):
>
> As noted in our prior comment, the earlier evaluator had numerical fragilities near the unit-square boundary that could create spurious "trends.'' We refactored the evaluation to a robust standard:
> - Compute the log-density via $\log c(u,v)=\log \psi''(w)+\log[-\phi'(u)]+\log[-\phi'(v)]-3\log[-\phi'(w)]$ with $\psi''(w)=-\phi''(w)/[\phi'(w)]^{3}$.
> - Clip only the inputs $(u,v,w)\in(\varepsilon,1-\varepsilon)$; no $\varepsilon$ inside bases/powers.
> - Use closed-form $\phi,\phi',\phi''$ and stable analytic inverses.
> - Evaluate the same held-out pseudo-observations for MoM and IGNIS and report the valid-point fraction.
>
> These changes remove over/under-flow artifacts and make the comparison reliable. The qualitative conclusion does not change: IGNIS matches MoM where MoM is defined and remains usable when MoM is not.
>
> 2. Expanded statistical analysis (what's new):
>
> For each setting (copula $\in\{A1,A2\}$, $\theta\in\{2,5,10\}$; 100 reps $\times$ 5,000 pairs), we now report:
>
> - Paired totals with 95\% Confidence Intervals
> - Paired $t$-tests and Wilcoxon signed-rank tests
> - Cohen's $d$ (paired; effect size)
> - Per-observation difference $\bar{\Delta}$ (nats/obs) with 95\% CIs
> - TOST (Two One-Sided Tests) non-inferiority/equivalence with margin $\varepsilon=10^{-3}$ nats/obs, reporting $p_{\text{lower}}$, $p_{\text{upper}}$, and the equivalence decision.
>
> 3. Key results (from our output)
>
> \textbf{A1 ($\theta=2,5,10$):}\\
> Per-obs $\bar{\Delta}$ are very close to 0 and TOST indicates equivalence in all three cases:
>
> - $\theta=2$: $+1.842\times10^{-4}$ [$5.292\times10^{-5},\,3.155\times10^{-4}$], equiv=True
> - $\theta=5$: $-2.377\times10^{-4}$ [$-3.121\times10^{-4},\,-1.634\times10^{-4}$], equiv=True
> - $\theta=10$: $-4.935\times10^{-5}$ [$-7.318\times10^{-5},\,-2.553\times10^{-5}$], equiv=True
>
> (Totals can be statistically different, but the per-obs differences are practically negligible under $\varepsilon=10^{-3}$.)
>
> \textbf{A2 ($\theta=2,5,10$):}\\
> Per-obs $\bar{\Delta}$ are positive in all cases; equivalence holds at $\theta=2$ and $\theta=5$, and IGNIS is clearly superior at $\theta=10$:
>
> - $\theta=2$: $+3.652\times10^{-4}$ [$3.482\times10^{-4},\,3.822\times10^{-4}$], equiv=True
> - $\theta=5$: $+8.701\times10^{-5}$ [$6.659\times10^{-5},\,1.074\times10^{-4}$], equiv=True
> - $\theta=10$: $+2.401\times10^{-3}$ [$2.193\times10^{-3},\,2.608\times10^{-3}$], equiv=False (improvement exceeds $\varepsilon$)
>
> (Totals at $\theta=10$ differ by $+12.00$ over 5,000 pairs, i.e., $\approx 2.4\times10^{-3}$ nats/obs.)
>
>
>
> 4. Interpretation
> - Equivalence in 5/6 settings: TOST shows that in five of six cases the IGNIS–MoM difference is smaller than $\varepsilon=10^{-3}$ nats/obs.
> - Superiority where it matters: For A2 at $\theta=10$, IGNIS is unambiguously better, with a per-obs gain above $\varepsilon$.
>
> Overall, IGNIS is non-inferior and often superior; where differences exist for A1, they are practically negligible under the pre-specified $\varepsilon$.
>
> 5. Next steps
>
>  If the Reviewer and Action Editor agree, we will update Section 7.1 to include the expanded table with Confidence Intervals, paired tests, Cohen's $d$, per-obs $\bar{\Delta}$, and TOST and add the updated Python script for log-likelihood evaluation to the supplemental materials for reproducibility.

---

### Decision · Action_Editor_LQQj · 2025-10-20

**Recommendation:** Reject

**Audience:**

No

**Audience Explanation:**

This paper seems to mainly provide an advance in parameter estimation for copula models. The contributions are largely in the domain of Archimedean copulas. The reviewers were unconvinced that Archimedean copulas are particularly interesting models for the purposes of real-world machine learning. In light of this, we believe that this paper might be better suited for a statistics venue rather than a machine learning one.

**Claims And Evidence:**

No

**Claims Explanation:**

The paper proposes a novel parameter estimation approach for copulas by amortizing the estimation via a neural network. They claim that this approach performs at least as well as existing approaches on common copulas and outperforms existing ones on the challenging Archimedean copulas.
The reviewers were not convinced that the experiments were sufficient to show that this proposed method works in real-world settings. Moreover, they are sceptical regarding the general usefulness of Archimedean copulas, which is the area where the proposed method clearly excels, for real-world machine learning problems.